# Towards biologically plausible phosphene simulation for the differentiable optimization of visual cortical prostheses

**Maureen van der Grinten**[1†], **Jaap de Ruyter van Steveninck**[2*†], **Antonio Lozano**[1†], **Laura Pijnacker**[2], **Bodo Rueckauer**[2], **Pieter Roelfsema**[1], **Marcel van Gerven**[2], **Richard van Wezel**[2,3], **Umut Güçlü**[2‡], **Yağmur Güçlütürk**[2‡]

[1]Netherlands Institute for Neuroscience, Vrije Universiteit, Amsterdam, Netherlands; [2]Donders Institute for Brain Cognition and Behaviour, Radboud University Nijmegen, Nijmegen, Netherlands; [3]Biomedical Signals and Systems Group, University of Twente, Enschede, Netherlands

**\*For correspondence:**
jaap.deruyter@donders.ru.nl

[†]These authors contributed equally to this work
[‡]These authors also contributed equally to this work

**Competing interest:** The authors declare that no competing interests exist.

**Abstract** Blindness affects millions of people around the world. A promising solution to restoring a form of vision for some individuals are cortical visual prostheses, which bypass part of the impaired visual pathway by converting camera input to electrical stimulation of the visual system. The artificially induced visual percept (a pattern of localized light flashes, or 'phosphenes') has limited resolution, and a great portion of the field's research is devoted to optimizing the efficacy, efficiency, and practical usefulness of the encoding of visual information. A commonly exploited method is non-invasive functional evaluation in sighted subjects or with computational models by using simulated prosthetic vision (SPV) pipelines. An important challenge in this approach is to balance enhanced perceptual realism, biologically plausibility, and real-time performance in the simulation of cortical prosthetic vision. We present a biologically plausible, PyTorch-based phosphene simulator that can run in real-time and uses differentiable operations to allow for gradient-based computational optimization of phosphene encoding models. The simulator integrates a wide range of clinical results with neurophysiological evidence in humans and non-human primates. The pipeline includes a model of the retinotopic organization and cortical magnification of the visual cortex. Moreover, the quantitative effects of stimulation parameters and temporal dynamics on phosphene characteristics are incorporated. Our results demonstrate the simulator's suitability for both computational applications such as end-to-end deep learning-based prosthetic vision optimization as well as behavioral experiments. The modular and open-source software provides a flexible simulation framework for computational, clinical, and behavioral neuroscientists working on visual neuroprosthetics.

## Editor's evaluation

This important study presents a simulator for prosthetic vision (with open source code) whose design is informed by previous psychophysical and neuroanatomical work. The simulation is convincing and demonstrates significant improvements over past visual prosthesis simulations. This work will be of interest to those investigating the impact of cortical stimulation on perception, particularly those developing visual prostheses.

## Introduction

Globally, as per 2020, an estimated 43.3 million people were blind (*Bourne et al., 2021*). For some cases of blindness, visual prosthetics may provide a promising solution. These devices aim to

restore a rudimentary form of vision by interacting with the visual system using electrical stimulation (*Fernandez, 2018*; *Bloch et al., 2019*; *Nowik et al., 2020*). In particular, our work concerns prosthetic devices that target the primary visual cortex (V1). Despite recent advances in the field, more research is required before cortical prosthesis will become clinically available. Besides research into the improvement of the safety and durability of cortical implants (*Chen et al., 2020*; *Fernández et al., 2021*), a great portion of the research attention is devoted to optimizing the efficacy, efficiency, and practical usefulness of the prosthetic percepts. The artificially induced visual percepts consist of patterns of localized light flashes ('phosphenes') with a limited resolution. To achieve a functional level of vision, scene-processing is required to condense complex visual information from the surroundings in an intelligible pattern of phosphenes (*de Ruyter van Steveninck et al., 2022a*; *Lozano et al., 2020*; *Normann et al., 2009*; *Sanchez-Garcia et al., 2020*; *Troyk, 2017*; *Granley et al., 2022a*; *Han et al., 2021*). Many studies employ a SPV paradigm to non-invasively evaluate the functional quality of the prosthetic vision with the help of sighted subjects (*de Ruyter van Steveninck et al., 2022b*; *Beyeler et al., 2017*; *Normann et al., 2009*; *Cha et al., 1992a*; *Dagnelie et al., 2006*; *Dagnelie et al., 2007*; *Han et al., 2021*; *Vergnieux et al., 2014*; *Srivastava et al., 2009*; *Cha et al., 1992b*; *Sommerhalder et al., 2004*; *Bollen et al., 2019*; *Winawer and Parvizi, 2016*; *Pezaris and Reid, 2008*; *Vurro et al., 2014*; *Rassia et al., 2022*) or through 'end-to-end' approaches, using in silico models (*de Ruyter van Steveninck et al., 2022a*; *Küçükoğlu et al., 2022*; *Granley et al., 2022a*). Although the aforementioned SPV literature has provided us with important insights, the perceptual realism of electrically generated phosphenes and some aspects of the biological plausibility of the simulations can be further improved by integrating knowledge of phosphene vision and its underlying physiology. Given the steadily expanding empirical literature on cortically-induced phosphene vision, it is both feasible and desirable to have a more phenomenologically accurate model of cortical prosthetic vision. Such an accurate simulator has already been developed for retinal prostheses (*Beyeler et al., 2017*), which has formed an inspiration for our work on simulation of cortical prosthetic vision. Thus, in this current work, we propose a realistic, biologically inspired computational model for the simulation of cortical prosthetic vision. Biological plausibility, in our work's context, points to the

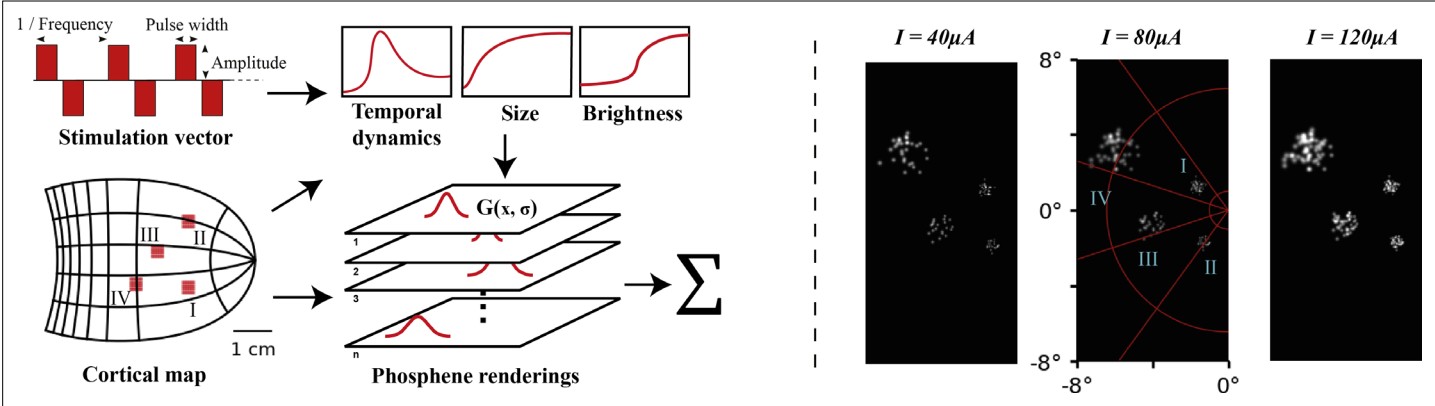

**Figure 1.** Left: schematic illustration of our simulator pipeline. Our simulator is initialized with electrode locations on a visuotopic map of the visual cortex. Each frame, the simulator takes a set of stimulation parameters that for each electrode specify the amplitude, pulse width, and frequency of electrical stimulation. Based on the electrode locations on the cortical map and the stimulation parameters, the phosphene characteristics are estimated and for each phosphene the effects are rendered on a map of the visual field. Finally, the phosphene renderings are summed to obtain the resulting simulated prosthetic percept. Right: Example renderings after initializing the simulator with four 10 × 10 electrode arrays (indicated with roman numerals) placed in the right hemisphere (electrode spacing: 0.4 mm, in correspondence with the commonly used 'Utah array' *Maynard et al., 1997*). The output is visualized for 166 ms pulse trains with stimulation amplitudes of 40, 80, and 120 μA, a pulse width of 170 ms, and a frequency of 300Hz. In these example frames, we can observe the effects of cortical magnification, thresholds for activation, current-dependent spread (size) and proportion (brightness) of cortical tissue activation.

The online version of this article includes the following figure supplement(s) for figure 1:

**Figure supplement 1.** Demonstration of the use of the simulator in conjunction with external software for receptive field prediction (*Goebel et al., 2006*).

**Figure supplement 2.** Example simulation of irregular phosphene shapes.

**Figure supplement 3.** Example simulations of irregular phosphene percepts that can arise upon multi-electrode stimulation of three electrodes.

simulation's ability to capture essential biological features of the visual system in a manner consistent with empirical findings: our simulator integrates quantitative findings and models from the literature on cortical stimulation in V1. The elements that are modeled in our simulator include cortical magnification, current-dependent spread of activation, and charge-dependent activation thresholds. Furthermore, our simulator models the effects of specific stimulation parameters, accounting for temporal dynamics. A schematic overview of the pipeline and some example outputs are displayed in *Figure 1*. Our simulator runs in real-time, is open-source, and makes use of fully differentiable functions, which is an essential requirement for the gradient-based optimization of phosphene encoding models with machine learning. This design enables both simulations with sighted participants, as well as end-to-end optimization in machine-learning frameworks, thus fitting the needs of fundamental, clinical, and computational scientists working on neuroprosthetic vision. All of the underlying models in the simulator are flexibly and modularly implementend allowing for easy integration with external software (see *Figure 1—figure supplement 1*). Although the simulator, by default, assumes round phosphene percepts that are independently activated, it can be tailored to simulate custom alternative scenarios: the phosphene maps can be adjusted to simulate different shapes (see *Figure 1—figure supplement 2*), and it is possible to simulate arbitrary interactions between electrodes with minor adaptations to our code (see *Figure 1—figure supplement 3*). The modular design of the simulator allows for future extensions to simulate brain stimulation in other regions such as lateral geniculate nucleus (LGN) or higher visual areas (*Murphey and Maunsell, 2007*; *Pezaris and Reid, 2007*; *Panetsos et al., 2011*; *Mirochnik and Pezaris, 2019*).

## Background and related work

### Cortical prostheses

Early attempts by Brindley and Lewin, and Dobelle successfully reported the ability to reliably induce the perception of phosphenes (described as localized round flashes of light) via electrical stimulation of the cortical surface (*Brindley and Lewin, 1968*; *Dobelle and Mladejovsky, 1974*). More recent preclinical studies demonstrate promising results concerning the safety and efficacy of long-term stimulation in the primary visual cortex, either via surface electrodes (*Niketeghad et al., 2020*; *Beauchamp et al., 2020*) or with intracortical electrodes (*Bak et al., 1990*; *Schmidt et al., 1996*; *Fernández et al., 2021*). Other studies that performed V1 stimulation in sighted subjects (*Winawer and Parvizi, 2016*; *Bosking et al., 2017*) and non-human primates (*Schiller et al., 2011*; *Chen et al., 2020*) have shown similar success. Some milestones include the implantation of over 1000 electrodes in a monkey's visual cortex (*Chen et al., 2020*), and the testing of a preliminary artificial vision system that presents visual information from the surroundings to a blind subject using a penetrating multi-electrode array in the visual cortex (*Fernández et al., 2021*). Taken together, the previous literature provides strong evidence for the clinical potential of cortical prostheses for the blind.

### Perceptual reports on cortical prosthetic vision

In our simulator, we integrate empirical findings and quantitative models from existing literature on electrical stimulation in the visual cortex. Stimulation in V1 with intracortical electrodes is estimated to activate tens to thousands of neurons (*Histed et al., 2009*), resulting in the perception of often 'featureless' white dots of light with a circular shape (*Brindley and Lewin, 1968*; *Bak et al., 1990*; *Schmidt et al., 1996*; *Niketeghad et al., 2020*; *Fernández et al., 2021*). Due to the cortical magnification (the foveal information is represented by a relatively large surface area in the visual cortex as a result of variation of retinal RF size) the size of the phosphene increases with its eccentricity (*Winawer and Parvizi, 2016*; *Bosking et al., 2017*). Furthermore, phosphene size, stimulation threshold (defined as the minimum current to produce a visible phosphene 50% of the time) and brightness depend on stimulation parameters such as the pulse width, train length, amplitude, and frequency of stimulation (*Bak et al., 1990*; *Schmidt et al., 1996*; *Winawer and Parvizi, 2016*; *Bosking et al., 2017*; *Niketeghad et al., 2020*; *Fernández et al., 2021*). To account for these effects, we integrated and adapted previously proposed quantitative models that estimate the charge-dependent activation level of cortical tissue (*Tehovnik and Slocum, 2007*; *Winawer and Parvizi, 2016*; *Bosking et al., 2017*; *Geddes, 2004*; *Bruce et al., 1999*; *Kim et al., 2017*). Furthermore, our simulator includes a model of the temporal dynamics, observed by *Schmidt et al., 1996*, accounting for response-attenuation after prolonged or repeated stimulation, as well as the delayed 'offset' of phosphene perception.

## Simulated prosthetic vision

A wide range of previous studies has employed SPV with sighted subjects to non-invasively investigate the usefulness of prosthetic vision in everyday tasks, such as mobility (*Cha et al., 1992a*; *Dagnelie et al., 2007*; *de Ruyter van Steveninck et al., 2022b*; *Han et al., 2021*; *Vergnieux et al., 2017*), hand-eye coordination (*Srivastava et al., 2009*), reading (*Cha et al., 1992a*; *Sommerhalder et al., 2004*), or face recognition (*Bollen et al., 2019*; *Winawer and Parvizi, 2016*). Several studies have examined the effect of the number of phosphenes, spacing between phosphenes and the visual angle over which the phosphenes are spread (e.g. *de Ruyter van Steveninck et al., 2022b*; *Thorn et al., 2020*; *Sanchez-Garcia et al., 2022*; *Srivastava et al., 2009*; *Parikh et al., 2013*). The results of these studies vary widely, which could be explained by the difference in the implemented tasks, or, more importantly, by the differences in the simulation of phosphene vision. The aforementioned studies used varying degrees of simplification of phosphene vision in their simulations. For instance, many included equally-sized phosphenes that were uniformly distributed over the visual field (informally referred to as the 'scoreboard model'). Furthermore, most studies assumed either full control over phosphene brightness or used binary levels of brightness (e.g. 'on' / 'off'), but did not provide a description of the associated electrical stimulation parameters. Several studies have explicitly made steps towards more realistic phosphene simulations, by taking into account cortical magnification or using visuotopic maps (*Wong et al., 2010*; *Li, 2013*; *Srivastava et al., 2009*; *Paraskevoudi and Pezaris, 2021*), simulating noise and electrode dropout (*Dagnelie et al., 2007*), or using varying levels of brightness (*Vergnieux et al., 2017*; *Sanchez-Garcia et al., 2022*; *Parikh et al., 2013*). However, no phosphene simulations have modeled temporal dynamics or provided a description of the parameters used for electrical stimulation. Some recent studies developed descriptive models of the phosphene size or brightness as a function of the stimulation parameters (*Winawer and Parvizi, 2016*; *Bosking et al., 2017*). Another very recent study has developed a deep-learning based model for predicting a realistic phosphene percept for single stimulating electrodes (*Granley et al., 2022b*). These studies have made important contributions to improve our understanding of the effects of different stimulation parameters. The present work builds on these previous insights to provide a full simulation model that can be used for the functional evaluation of cortical visual prosthetic systems. Meanwhile, a realistic and biologically-plausible simulator has been developed for retinal prosthetic vision (Pulse2Percept, *Beyeler et al., 2017*), which takes into account the axonal spread of activation along ganglion cells and temporal nonlinearities to construct plausible simulations of stimulation patterns. Even though scientists increasingly realize that more realistic models of phosphene vision are required to narrow the gap between simulations and reality (*Han et al., 2021*; *Dagnelie, 2008*), a biophysically-grounded simulation model for the functional evaluation of cortical prosthetic vision remains to be developed. Realistic SPV can aid technological developments by allowing neuroscientists, clinicians, and engineers to test the perceptual effects of changes in stimulation protocols, and subsequently select stimulation parameters that yield the desired phosphene percepts without the need for extensive testing in blind volunteers. A realistic simulator could also be used as support in the rehabilitation process, assisting clinicians and caregivers in identifying potential problematic situations and adapt preprocessing or stimulation protocols accordingly (*Dagnelie, 2008*).

## Deep learning-based optimization of prosthetic vision

SPV is often employed to develop, optimize, and test encoding strategies for capturing our complex visual surroundings in an informative phosphene representation. Numerous scene-processing methods have been proposed in the literature, ranging from basic edge detection, or contour-detection algorithms (*Boyle et al., 2001*; *Dowling et al., 2004*; *Guo et al., 2018*) to more intelligent deep learning-based approaches, which can be tailored to meet task-specific demands (*Sanchez-Garcia et al., 2020*; *Han et al., 2021*; *Rasla and Beyeler, 2022*; *Bollen et al., 2019*; *de Ruyter van Steveninck et al., 2022a*; *Lozano et al., 2020*; *Lozano et al., 2018*). The advantages of deep learning-based methods are clear: more intelligent and flexible extraction of useful information in camera input leads to less noise or unimportant information in the low-resolution phosphene representation, allowing for more successful completion of tasks. Some recent studies demonstrated that the simulation of prosthetic vision can even be incorporated directly into the deep learning-based optimization pipeline for end-to-end optimization (*de Ruyter van Steveninck et al., 2022a*; *Küçükoğlu et al., 2022*; *Granley et al., 2022a*). With end-to-end optimization, the image processing can be tailored to an individual user or

specific tasks. Here, the usefulness of the prosthetic vision is evaluated by a computational agent, or decoder model, which assists in the direct optimization of the stimulation parameters required to optimally encode the present image. Again, an important drawback is that the current computational studies use simplified simulations of cortical prosthetic vision. This problem is addressed in the current study. Note that end-to-end machine learning pipelines rely on gradient propagation to update the parameters of the phosphene encoding model. Consequently, a crucial requirement is that the simulator makes use of differentiable operations to convert the stimulation parameters to an image of phosphenes - a requirement that is met by the proposed simulator. To evaluate the practical usability of our simulator in an end-to-end framework, we replicated and adapted the experiments by *de Ruyter van Steveninck et al., 2022a*, replacing their simulator with ours. The currently proposed simulator can be compared to the simulator that was used in the aforementioned work: our simulator can handle temporal sequences and our experiments explore a more biologically grounded simulation of phosphene size and locations. Furthermore, instead of a more abstract or qualitative description of the required stimulation ('on' / 'off'), we included biologically inspired methods to model the perceptual effects of different stimulation parameters such as the current amplitude, the duration, the pulse width, and the frequency. This opens new doors for optimization of the stimulation parameters in realistic ranges: although technological developments advance the state-of-the-art hardware capabilities rapidly, cortical prosthesis devices will be operating under energy constraints, due to both hardware limitations as well as safety limits regarding neurostimulation (*Shannon, 1992*; *McCreery et al., 2002*). Deep learning methods trained in tandem with a biologically plausible phosphene simulator can be leveraged to produce constrained optimal stimulation paradigms that take these limitations into account, allowing for safe and viable stimulation protocols to be developed.

## Materials and methods

Our simulator is implemented in Python, using the PyTorch deep learning library (*Paszke et al., 2019*). The simulator makes use of differentiable functions which, given the entire set of phosphenes and their modeled properties, calculate the brightness of each pixel in the output image in parallel. This architecture makes our model memory intensive, but allows for fast computations that can be executed on a GPU. Each frame, the simulator maps electrical stimulation parameters (stimulation current, pulse width, and frequency) to an estimated phosphene perception, taking into account the stimulation history. In the sections below, we discuss the different components of the simulator model, followed by a description of some showcase experiments that assess the ability to fit clinical data and the practical usability of our simulator in simulation experiments. Our simulator can be imported as a python package and the source code is available on GitHub. The source code of our simulator can be retrieved from GitHub (copy archived at *de Ruyter van Steveninck, 2024a*). The latest stable release can be installed using pip:

```
$ pip install dynaphos.
```

### Visuotopic mapping

Our simulator can be flexibly initialized with a list of electrode locations or phosphene locations to the simulator to base the simulations on clinical data. We also provide the code for generating a random list of phosphene locations based on equally-distant electrode locations on a flattened cortical map of V1. For mapping phosphene locations from the cortical electrode locations to the neuroprosthesis user's visual field, our simulator uses the reverse wedge-dipole visuotopic model of V1, proposed by *Polimeni et al., 2006*. This model maps a complex polar coordinate $z = re^{i\theta}$ in the visual field, to a cortical location $w$ in one hemisphere of V1, following the visuotopic relationship

$$w = k \left( \log(re^{i\alpha\theta} + a) - \log(re^{i\alpha\theta} + b) \right) .$$

(1)

Here, $r$ and $-\frac{\pi}{2} \leq \theta \leq \frac{\pi}{2}$ are the eccentricity and azimuth of the point in the visual field, the parameter $\alpha$ controls the shear of the 'wedge map', $k$ is a scaling factor that scales the mapping to realistic proportions in cortical distance (millimeters), and $a$ and $b$ are parameters that control the singularities of the dipole model. For the mapping from cortical coordinates to phosphene location, we use the inverse of *Equation 1*, which is given by

$$z = \Lambda^{-1} \left( \frac{ab\left(e^{\frac{w}{k}} - 1\right)}{b - ae^{\frac{w}{k}}} \right) \tag{2}$$

for the inverse shearing operation

$$\Lambda^{-1}\left(re^{i\theta}\right) = re^{i\frac{\theta}{\alpha}} . \tag{3}$$

The visuotopic model also provides us with the cortical magnification $M$, which defines the relative amount of cortical tissue that is involved in processing of visual information, depending on the eccentricity in the visual field. The cortical magnification is given by the derivative of *Equation 1* along the horizontal meridian:

$$M = \frac{k\left(b - a\right)}{\left(r + a\right)\left(r + b\right)} . \tag{4}$$

Here, $M$ is given in millimetres of cortical surface per degree of visual angle. The parameters of the models are configurable. The default values are specified below, in the 'Parameter estimates' section. Note that in our simulation software, we provide the option of substituting *Equations 1, 2, 3 and 4*, with other estimates described such as the mono- or dipole model in *Polimeni et al., 2006*; *Schwartz, 1983*. Moreover, to simulate imperfect knowledge of electrode or phosphene locations, and malfunctioning electrodes, the cortical mapping methods include parameters for the introduction of noise and electrode dropout. We note, however, that the framework is compatible with the retinotopic maps of other structures, such as the LGN, which is the structure providing input to the primary visual cortex.

## Phosphene size

Based on a model by *Tehovnik and Slocum, 2007*, the phosphene size (in degrees),

$$P = \frac{D}{M} \tag{5}$$

is obtained via an estimation of the current spread from the stimulating electrodes, where

$$D = 2\sqrt{\frac{I}{K}} \tag{6}$$

is the diameter of the activated cortical tissue (in mm), for stimulation current $I$ (in µA) and excitability constant $K$ (in $\mathrm{A\,mm^{-2}}$). Note that the cortical magnification factor $M$ is obtained in *Equation 4*. The default value for $K$ is specified below, in the 'Parameter estimates' section. In our simulation software, we provide the option to substitute *Equation 6* with an estimate by *Bosking et al., 2017*. Based on verbal descriptions (*Schmidt et al., 1996*; *Bak et al., 1990*; *Fernández et al., 2021*), phosphenes are shown as Gaussian blobs with two standard deviations set equal to the phosphene size $P$, such that 95% of the Gaussian falls within the fitted phosphene size.

## Phosphene brightness

The brightness and detection threshold of each phosphene are based on a model of the intracortical tissue activation in response to electrical stimulation with biphasic square pulse trains. The model assumes brightness and detection thresholds of phosphene perception to be primarily correlated with the deposited charge, and accounts for the relative inefficiency of higher stimulation frequencies, longer pulse widths, and longer train durations, as found in *Winawer and Parvizi, 2016*; *Niketeghad et al., 2020*; *Fernández et al., 2021*. We model the combined effects of these stimulation parameters as follows: First, we subtract from the stimulation amplitude $I_{\mathrm{stim}}$ a leak current $I_0$, which represents the ineffective component of the stimulation input, and a memory trace $B$ (further explained in section 'Temporal dynamics') that accounts for the decreased neural excitability after prior stimulation. $I_0$ is set equal to the rheobase current (the absolute threshold for continuous stimulation at infinite

duration), following prior literature on the strength-duration relationship of neural tissue activation for isolated single-pulse trials (*Geddes, 2004*). To calculate the effective stimulation current of trains of pulses, the remaining current amplitude is multiplied with the duty cycle of the stimulation signal ($Pw \cdot f$, the fraction of one period in which the signal is active), such that

$$I_{\text{eff}} = \max\left(0, \quad \left(I_{\text{stim}} - I_0 - B\right) \cdot Pw \cdot f\right) \tag{7}$$

for pulse width $Pw$ and frequency $f$. Then, the cortical tissue activation is estimated by integrating the effective input current over multiple frames, using a leaky integrator model. By integrating over time, this model additionally implements the delayed on- and offset as described by several studies (*Schmidt et al., 1996*; *Bak et al., 1990*). For each frame with duration $\Delta t$, the estimated cortical activation is updated as

$$A_t = A_{t-\Delta t} + \Delta A \tag{8}$$

with

$$\Delta A = \left(-\frac{A_{t-\Delta t}}{\tau_{\text{act}}} + I_{\text{eff}} \cdot d\right) \cdot \Delta t. \tag{9}$$

Here, $\tau_{\text{act}}$ is the time constant of the activation decay in seconds and $d \in (0, 1]$ is a parameter that scales the duration of the stimulation relative to the frame duration. By default, $d$ is set to 1 to simulate a stimulation duration equal to the frame duration, where the total pulse train duration is controlled with the number of successive frames in which stimulation is provided to the simulator. Finally, if the cortical activation reaches the detection threshold (explained in the following section), the phosphene is activated with a brightness equal to the sigmoidal activation

$$\frac{1}{1 + e^{-\lambda(A - A_{50})}}, \tag{10}$$

where $\lambda$ is the slope of the sigmoidal curve and $A_{50}$ is the value of $A$ for which the phosphene reaches half the maximum brightness.

## Stimulation threshold

Our simulator uses a thresholding model based on psychometric data from *Fernández et al., 2021*. Phosphenes are only generated when the cortical tissue activation reaches the activation threshold $A_{\text{thr}}$, which is obtained for each electrode separately upon initialization of the simulator. To introduce a degree of variability between electrodes, $A_{\text{thr}}$ is sampled from the normal distribution

$$\mathcal{N}(\theta_{50}, \sigma^2). \tag{11}$$

The default values of the 50% probability threshold $\theta_{50}$, and the standard deviation $\sigma$ are fit on data from *Fernández et al., 2021* and can be found below, in the 'Parameter estimates' section. Note that, by default, the detection thresholds remain constant after initialization. However, in accordance to the user requirements, the values can be flexibly adjusted or re-initialized manually.

## Temporal dynamics

Using a memory trace of the stimulation history, the simulator accounts for basic accommodation effects on brightness for prolonged or repeated stimulation, as described in prior work (*Schmidt et al., 1996*). Each frame, the memory trace is dynamically updated as follows:

$$B_t = B_{t-\Delta t} + \Delta B \tag{12}$$

with

$$\Delta B = \left(-\frac{B_{t-\Delta t}}{\tau_{\text{trace}}} + I_{\text{eff}} \cdot \kappa\right) \Delta t. \tag{13}$$

Here, $\tau_{\text{trace}}$ is the time constant of the trace decay in seconds, and the parameter $\kappa$ controls the input effect. Note that the memory trace is used for the phosphene brightness and not the phosphene size. Because there is little experimental data on the temporal dynamics of phosphene size in relation to the accumulated charge, only the instantaneous current is used in the calculation of the phosphene size.

## Parameter estimates

By default, our model uses the parameters specified below. Unless stated otherwise, these parameter estimates were obtained by fitting our model to experimental data using the SciPy Python package, version 1.9.0 (*Virtanen et al., 2020*). More details on the comparison between the models' estimates and the experimental data can be found in the Results section. Note that the parameter settings may strongly depend on the specific experimental conditions (such as the type of electrodes).

- In *Equations 1, 2, 3 and 4* we use $a = 0.75$, $k = 17.3$, $b = 120$, and $\alpha = 0.95$, based on a fit by *Polimeni et al., 2006* on data of the human V1 from *Horton, 1991*.
- In *Equation 6*, the parameter $K$ is set to 675 μA mm$^{-2}$, following an estimate by *Tehovnik et al., 2006*, who measured the responses to intracortical stimulation in V1 at different current levels.
- In *Equation 7*, we use a rheobase current $I_0$ of 23.9 μA based on a fit on data from *Fernández et al., 2021*. Here, we used the strength-duration curve for tissue-excitability $Q_{\text{thr}} = I_0(c + t)$, with minimal input charge $Q_{\text{thr}}$, chronaxie parameter $c$ and total stimulation duration $t$ as described in *Geddes, 2004*.
- In *Equation 9*, the parameter $d$ is set equal to 1. The parameter $\tau_{\text{act}}$ is set equal to 0.111 s to reflect qualitative descriptions found in *Schmidt et al., 1996*. Note: this parameter is not obtained by fitting to experimental data.
- In *Equation 10*, we use a slope $\lambda = 19.2 \cdot 10^7$ and offset $A_{50} = 1.06 \cdot 10^{-6}$ for the brightness curve, based on a fit of our model on data by *Fernández et al., 2021*.
- The descriptive parameters of the distribution 11, are set to $\theta_{50} = 9.14 \cdot 10^{-8}$ and $\sigma = 6.72 \cdot 10^{-8}$, based on a fit on psychometric data by *Fernández et al., 2021*.
- In *Equations 12 and 13*, we use $\tau_{\text{trace}}$ set to $1.97 \times 10^3$ s and $\kappa$ set to 14.0. These values are based on a fit of our model to data from *Schmidt et al., 1996*.

## Results

In this section, we present the results of computational experiments and comparisons with the literature to verify and validate the biological realism, the performance, and the practical usability of the simulator.

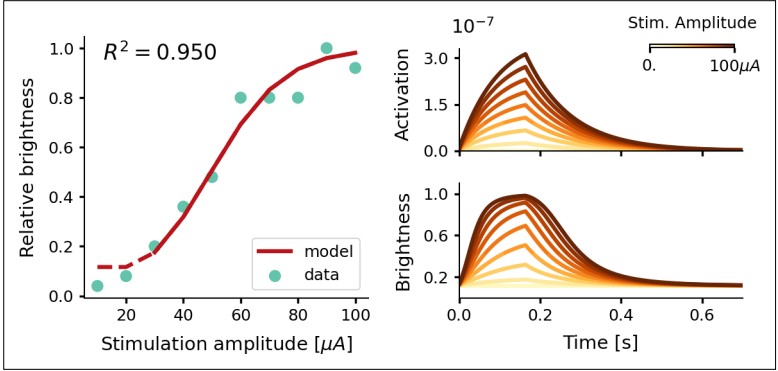

**Figure 2.** Estimate of the relative phosphene brightness for different stimulation amplitudes. The simulator was provided with a stimulation train of 166 ms with a pulse width of 170 μs at a frequency of 300 Hz (see *Equations 7, 8, 9 and 10*). Left: the fitted brightness levels reproduced by our model (red) and psychometric data reported by *Fernández et al., 2021* (light blue). Note that for stimulation amplitudes of 20 μA and lower, the simulator generated no phosphenes as the threshold for activation was not reached. Right: the modeled tissue activation and brightness response over time.

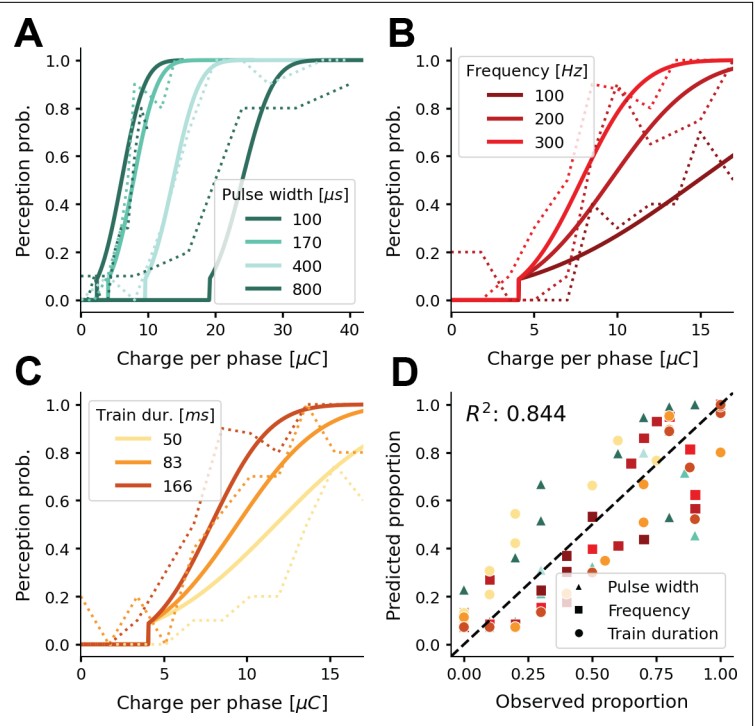

**Figure 3.** Probability of phosphene perception for different stimulation parameters. (**A–C**) Psychometric curves (solid lines) overlaid on experimental data (dashed lines) (*Fernández et al., 2021*; *Figure 2a and b*). The model's probability of phosphene perception is visualized as a function of charge per phase for (**A**) different pulse widths, (**B**) different frequencies, and (**C**) different train durations. Note that rather than the total charge per trial, we report the charge per phase to facilitate easy comparison with aforementioned experimental data. In panel (**D**) the probabilities of phosphene perception reproduced with our model are compared to the detection probabilities reported in (*Fernández et al., 2021*; *Figure 2a and b*). Predicted probabilities in panel (D) are the results of a threefold cross-validation on held-out test data. Colors conform to the conditions in panels A, B, and C.

The online version of this article includes the following figure supplement(s) for figure 3:

**Figure supplement 1.** Sixfold cross-validation results of fitting the simulator to data from several clinical studies that used cortical surface electrodes (*Dobelle and Mladejovsky, 1974*; *Girvin et al., 1979*; *Niketeghad et al., 2020*).

**Figure supplement 2.** Demonstration of the effect of the model's temporal dynamics over time.

## Biological plausibility

Here, we report on experimental data obtained from the literature, and evaluate the capacity of our simulator of fitting these empirical data. Using *Equations 7, 8, 9 and 10*, the simulator accurately reproduces the relative phosphene brightness that was reported in the previous study for different stimulation amplitudes ($R^2 = 0.950$; verification on the same data that was used to fit the model; *Figure 2*). *Figure 3* visualizes the effect of changing the stimulation parameters on the probability of phosphene perception, as estimated by our model. We compare our estimates with data reported by *Fernández et al., 2021*. The model accurately fits the reported effect of pulse width, frequency and train duration on the probability of phosphenes perception. To evaluate the robustness of this fit, a threefold cross-validation was performed (*Figure 3* panel (d)) where part of the data were held-out (i.e. the data in this panel were predicted and not in the data set used for the fit). In this more strict analysis, the prediction performance is still accurate (average $R^2 = 0.844$). *Figure 3—figure supplement 1* (not part of the main analysis) displays the cross-validation results after fitting the simulator to thresholding data from clinical studies that used cortical surface electrodes (*Dobelle and Mladejovsky, 1974*; *Girvin et al., 1979*; *Niketeghad et al., 2020*).

*Figure 4* displays the simulator's fit on the temporal dynamics found in a previous published study by *Schmidt et al., 1996*. Here, cross-validation was not feasible due to the limited amount of quantitative data. For repeated stimulation at different timescales (intervals of 4 s, and intervals of 200 s),

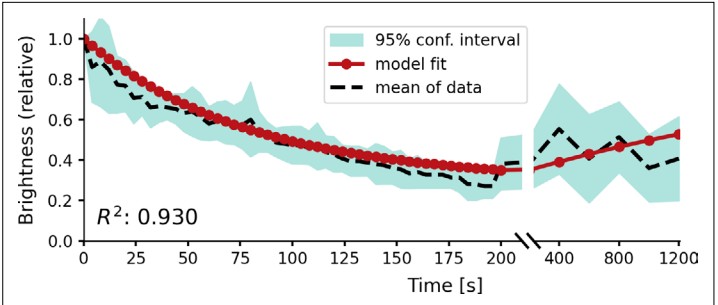

**Figure 4.** Relative brightness of a phosphene in response to repeated stimulation, overlaid on experimental results by *Schmidt et al., 1996*. The stimulation sequence consisted of 50 pulse trains at a 4 s stimulation interval, followed by five pulse trains at an interval of 200 s to test recovery. The simulator was provided with a stimulation train of 125 ms with a pulse width of 100 μs at a frequency of 200 Hz using a stimulation amplitude of 90 μA. Please notice the split x-axis with variable scaling.

the brightness of a single phosphene is evaluated after fitting the memory trace parameters. The observed accommodation effects in the simulator are compared to the data from *Schmidt et al., 1996*. *Figure 3—figure supplement 2* shows the effect of the modeled temporal dynamics on the simulator's output for continuous stimulation over 7 s.

## Performance

We tested the computational efficiency of our simulator, by converting a pre-processed example video (1504 frames) into simulated phosphene images, for different numbers of phosphenes, and at varying image resolutions. The used example video can be downloaded via this link. The simulator was run on a CUDA-enabled graphics card (NVIDIA A30) and each setting was run five times. The results are displayed in *Figure 5*. The lowest measured frame rate (10.000 phosphenes at a resolution of 256 × 256) was 28.7 frames per second. Note that the missing combinations in *Figure 5* indicate that the required memory exceeded the capacity of our GPU, as the simulation of large numbers of phosphenes at high resolutions can be memory intensive. Notably, even on a consumer-grade GPU (e.g. a 2016 model GeForce GTX 1080) the simulator still reaches real-time processing speeds (>100 fps) for simulations with 1000 phosphenes at 256 × 256 resolution. Some additional example videos are included in the online version of this article (*Videos 1 and 2*).

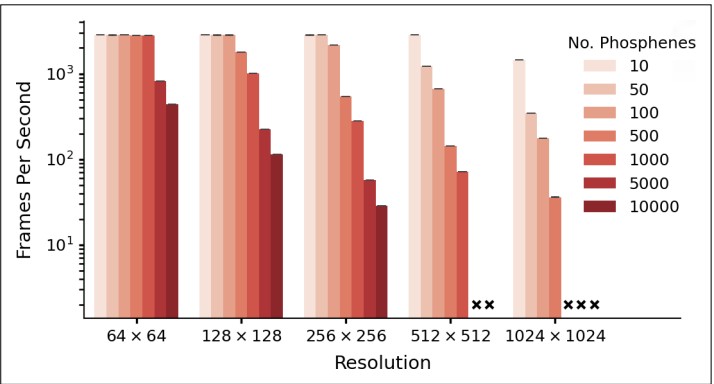

**Figure 5.** Performance as a function of resolution and number of phosphenes. The data is based on five runs of 1540 frames per condition, with batch size equal to one frame. Simulation was run with an NVIDIA A30 GPU (memory size: 24 GB). Crosses indicate missing conditions. Note that these data are presented only for evaluating the software-performance. For some combinations of phosphene count and image resolution (e.g. 10.000 phosphenes in a 64 × 64 image) there are fewer pixels than phosphenes. The error bars indicate the 95-percent confidence interval.

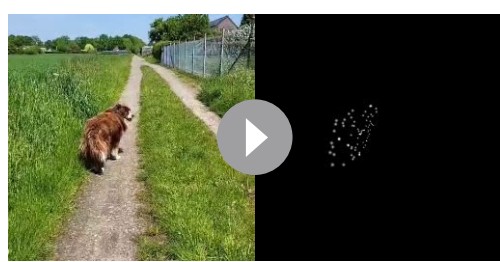

**Video 1.** Example phosphene simulation.
https://elifesciences.org/articles/85812/figures#video1

## Usability in a deep learning SPV pipeline

To validate that the simulator can conveniently be incorporated in a machine learning pipeline, we replicated an existing SPV pipeline by *de Ruyter van Steveninck et al., 2022a*, replacing the simulator of that study with our biologically plausible simulator. We performed several phosphene encoding optimization experiments, described below. In this pipeline, a convolutional neural network encoder is trained to process images or video frames and generate adequate electrode stimulation parameters. To train the encoder, a simulation of the prosthetic percept is generated by the phosphene simulator. This simulated percept is evaluated by a second convolutional neural network, the decoder, which decodes the simulated percept into a reconstruction of the original input image (*Figure 6*). The quality of the phosphene encoding is optimized by iteratively updating the network parameters of the encoder and decoder (simultaneously) using backpropagation of the reconstruction error. In addition to the reconstruction error, which measures similarity between the reconstruction and the input, we used a regularization term that measures similarity between the phosphenes and the input. For a more detailed description of the end-to-end optimization pipeline, see *de Ruyter van Steveninck et al., 2022a*.

### Dynamic end-to-end encoding of videos

In the first experiment, we explored the potential of using our simulator in a dynamic end-to-end encoding pipeline. The simulator is initialized with 1000 possible phosphenes in both hemifields, covering a field of view of 16° of visual angle. Note that the simulated electrode density and placement differs from current prototype implants and the simulation can be considered to be an ambitious scenario from a surgical point of view, given the folding of the visual cortex and the part of the retinotopic map in V1 that is buried in the calcarine sulcus. We extended the previously published pipeline with 3D-convolutions (with an additional temporal dimension) to enable encoding of subsequent video frames. The model was trained on a basic video dataset with moving white digits on a black background (Moving MNIST Dataset, *Srivastava et al., 2015*). We used video sequences of five frames. The framerate of the simulation was set at five frames per second. We used a combination of two equally-weighted mean squared error (MSE) loss functions: the MSE loss between reconstruction and input, and the MSE loss between the simulated phosphene representation and the input. *Figure 7* displays several frames after training for 45 epochs (for a total of 810,000 training examples). We can observe that the model has successfully learned to represent the original input frames in phosphene vision over time, and the decoder is able to approximately reconstruct the original input.

### Constrained stimulation and naturalistic scenes

In a second experiment, we trained the end-to-end model with a more challenging dataset containing complex images of naturalistic scenes (the ADE20K dataset *Zhou et al., 2019*). In this experiment, we implemented the original pipeline described in *de Ruyter van Steveninck et al., 2022a* (experiment 4), with the same phosphene coverage as the previously described experiment. The images were normalized and converted to grayscale, and we applied a circular mask such that the corners (outside the field covered by phosphenes) were ignored in the reconstruction task. The experiment consisted of three training runs, in which we tested different conditions: a free optimization condition, a constrained optimization condition, and a supervised boundary reconstruction condition. In the free optimization condition, the model was trained using an equally weighted combination of a MSE reconstruction loss between input and reconstruction, and a MSE regularization loss between the phosphenes and input images. After

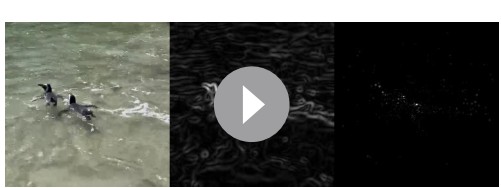

**Video 2.** Example phosphene simulation.
https://elifesciences.org/articles/85812/figures#video2

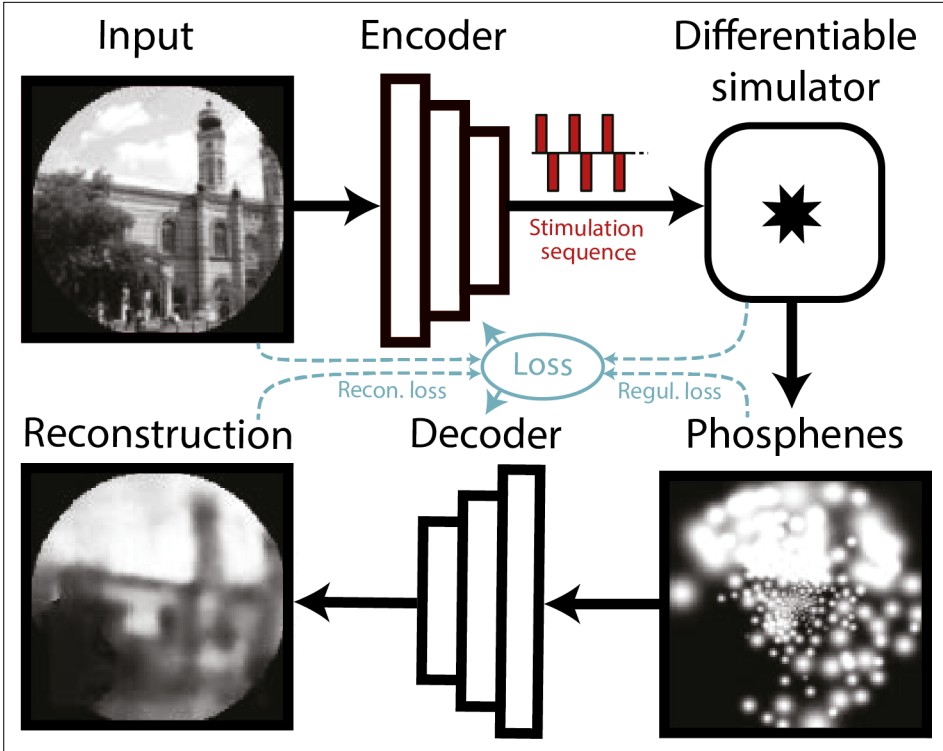

**Figure 6.** Schematic illustration of the end-to-end machine-learning pipeline adapted from *de Ruyter van Steveninck et al., 2022a*. A convolutional neural network encoder is trained to convert input images or video frames into a suitable electrical stimulation protocol. In the training procedure, the simulator generates a simulation of the expected prosthetic percept, which is evaluated by a second convolutional neural network that decodes a reconstruction of the input image. The quality of the encoding is iteratively optimized by updating the network parameters using back-propagation. Different loss terms can be used to constrain the phosphene encoding, such as the reconstruction error between the reconstruction and the input, a regularization loss between the phosphenes and the input, or a supervised loss term between the reconstructions and some ground-truth labeled data (not depicted here). Note that the internal parameters of the simulator (e.g. the estimated tissue activation) can also be used as loss terms.

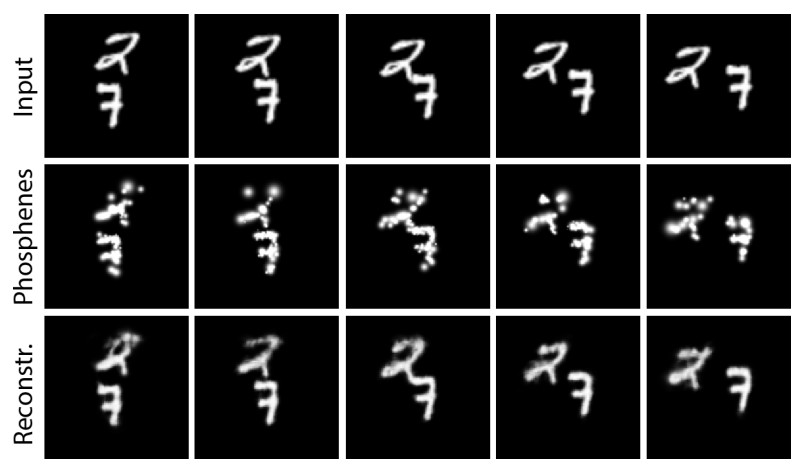

**Figure 7.** Results of training the end-to-end pipeline on video sequences from the moving MNIST dataset (*Srivastava et al., 2015*). Columns indicate different frames. Top row: the input frames; middle row: the simulated phosphene representations; bottom row: the decoded reconstructions of the input.

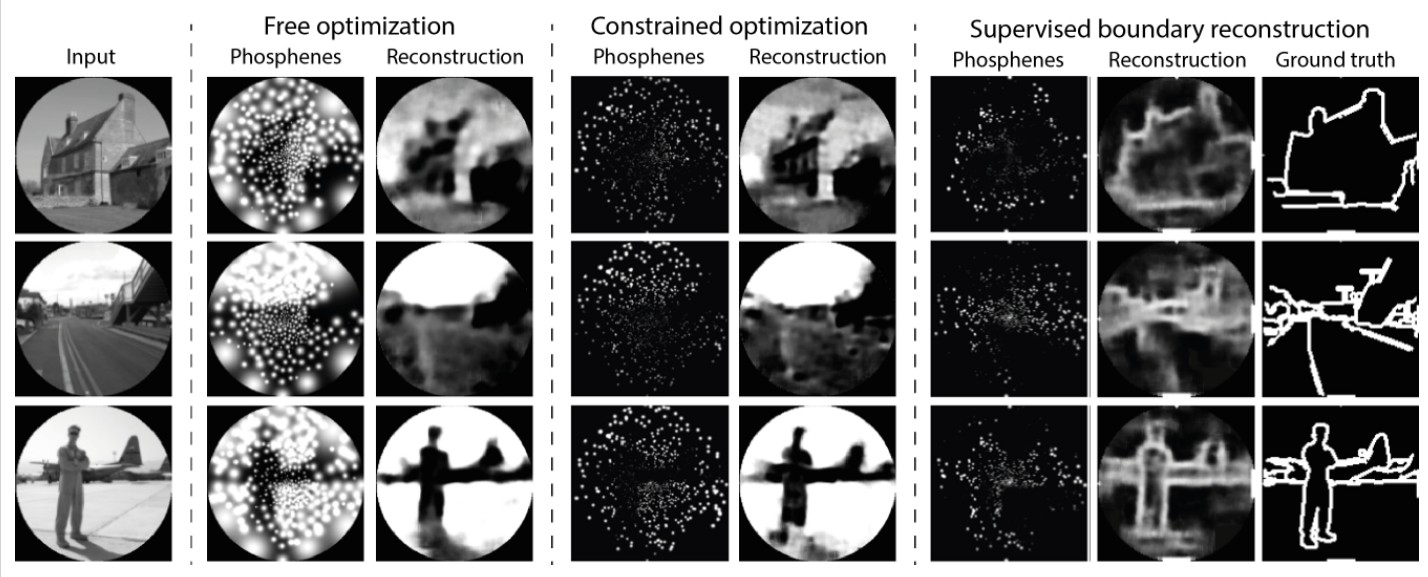

**Figure 8.** Results of training our simulator in an end-to-end pipeline on naturalistic images from the ADE20K dataset (*Zhou et al., 2019*). In the constrained optimization condition and the supervised boundary reconstruction condition, the encoder was configured to output 10 discrete stimulation amplitudes within the safe range of stimulation (0 to 128 µA). The selected images represent the first three categories in the validation dataset ('Abbey', 'Access Road,' 'Airbase'). Note that the brightness is enhanced in the phosphene images of the constrained optimization and the supervised boundary condition by 40%. (See *Figure 8—figure supplement 1* and *Figure 8—figure supplement 2* for enlarged and inverted simulated prosthetic vision (SPV) images).

The online version of this article includes the following figure supplement(s) for figure 8:

**Figure supplement 1.** Enlarged version of the simulated prosthetic vision (SPV) representations in *Figure 8*.

**Figure supplement 2.** Inverted grayscale version of the simulated prosthetic vision (SPV) representations in *Figure 8*.

**Figure supplement 3.** Example results after training an end-to-end model with a small electrode count.

**Figure supplement 4.** Results after training an end-to-end model on a small electrode count (60 electrodes) with different interaction conditions.

six epochs the model found an optimal encoding strategy that can accurately represent the scene and allows the decoder to accurately reconstruct pixel intensities while qualitatively maintaining the image structure (see *Figure 8*). Importantly, the encoder encoded brighter areas of the input picture with large stimulation amplitudes (over 2000 µA). The encoding strategy found in such an unconstrained optimization scheme is not feasible for real-life applications. In practice, the electrical stimulation protocol will need to satisfy safety bounds and it will need to comply with technical requirements and limitations of the stimulator hardware. For instance, rather than continuous stimulation intensities it is likely that the stimulator will allow for stimulation with only a number of (discrete) amplitudes. To evaluate whether our end-to-end pipeline can be harnessed to optimize the encoding in a constrained context, we performed a second training run (the constrained condition) where we reconfigured the encoder to output 10 discrete values between 0 and 128 µA. We used straight-through estimation with a smooth staircase function to estimate the gradients during backpropagation. To compensate for the relative sparsity of phosphenes in the SPV representation, we increased the training stability by taking the regularization loss as the MSE between the pixel brightness at the phosphene centers and the corresponding pixel brightness in the input image. Furthermore, to encourage large spatial correspondence with input stimuli, we adapted the relative weights of the reconstruction loss and the regularization loss to 0.00001 and 0.99999, respectively. Note that the regularization loss only promotes similarity between the phosphene encoding and the input and the decoder is unaffected by the regularization loss. The results of the safety-constrained training after six epochs are visualized in *Figure 8*. Note that overall, the resulting phosphenes are less bright and smaller due to the lower stimulation amplitudes. Nevertheless, the decoder is able to accurately reconstruct the original input. One limitation is that we did not test the subjective interpretability for human observers. As not all information in the scene is equally important, it may be informative to further constrain the phosphene representation to encode specific task-relevant features. In a third training run (the

supervised boundary condition) we validated whether our simulator can be used in a supervised machine learning pipeline for the reconstruction of specific target features, such as the object boundaries. Instead of using the input image as a reference, now the MSE is used between the reconstruction and a ground truth target image, and between the pixel brightness at the phosphene centers and the corresponding pixel brightness in the target image. The ground truth semantic boundary targets were obtained by performing canny edge detection and subsequent line thickening on the semantic segmentation labels provided with the dataset. The results after training for 16 epochs are visualized in *Figure 8*. Note that the model successfully converged to a sparse phosphene encoding that selectively represents the object boundaries. Enlarged and inverted visualizations can be found in *Figure 8—figure supplements 1 and 2*.

## Small electrode counts and interactions

The previously described experiments simulate implant designs with many electrodes. As a supplementary experiment, we verified that the end-to-end model can also be trained with smaller phosphene counts. The simulator was initialized with a random subset of 60 electrodes from a 10 × 10 array with electrode spacing 0.4 mm, matching the location and spacing of array no. IV in *Figure 1*. An image dataset was used with white characters on a black background. We tested three conditions to address potential interaction effects between neighboring electrodes. Besides a baseline training, we evaluated training the model with an additional loss component to avoid unexpected interactions by discouraging simultaneous activation of neighboring electrode pairs. This co-stimulation loss component is defined as

$$L_{\text{costim}} = \frac{1}{n}\sum_{i=1}^{n}\sum_{j=1}^{m}\left(\frac{I_i \cdot I_j}{1 + \|p_j - p_i\|^2}\right), \tag{14}$$

for stimulation currents $I_i$ and $I_j$, and $\|p_j - p_i\|^2$ the squared distance between the electrodes in mm. In a final training condition, instead of avoiding interactions using a loss component, we explicitly included an interaction model in the phosphene simulation. An electrode coactivation interaction was implemented, where current of active electrodes 'leaks' to neighboring activated electrodes based on their distance. For each active electrode pair $i$ and $j$ we added the coactivation current

$$I_{\text{coact}} = \sum_{j=1}^{m}\left(\frac{I_j}{1 + 100\|p_j - p_i\|^2}\right) \tag{15}$$

to the stimulation current $I_i$ used in the simulation. The results are visualized in *Figure 8—figure supplements 3 and 4*. The found encoding strategy resulted in distinct letter shapes. The letters are, however, poorly recognizable which is unsurprising with the minimal electrode resolution. The costimulation loss resulted in a lower percentage of active neighboring electrodes (at a distance < 1 mm). The electrode coactivation resulted in a higher percentage of active neighboring electrodes compared to the baseline, suggesting that the encoder learns to make use of the leak current.

## Discussion

The aim of this study is to present a biologically plausible phosphene simulator, which takes realistic ranges of stimulation parameters, and generates a phenomenologically accurate representation of phosphene vision using differentiable functions. In order to achieve this, we have modeled and incorporated an extensive body of work regarding the psychophysics of phosphene perception. The results indicate that our simulator is able to produce phosphene percepts that match the descriptions of phosphene vision that were gathered in basic and clinical visual neuroprosthetics studies over the past decades. When we used a GPU, the simulator ran in real-time, and as such, it could be used in experiments with sighted volunteers. Furthermore, our proof-of-principle computational experiments demonstrate the suitability of the simulator for machine learning pipelines, aimed at improving the image processing and stimulation strategies. Here, we discuss some implications of our findings.

## Validation experiments

### Visuotopic mapping

The results presented in *Figure 1* illustrate the value of including a visuotopic model based on the spread of cortical activation to realistically estimate phosphene locations and size. Some previous studies have used a model of cortical magnification (*Srivastava et al., 2009*; *Paraskevoudi and Pezaris, 2021*) or visuotopic mapping (*Wong et al., 2010*; *Li, 2013*) in their phosphene simulations. However, our simulator is the first to incorporate empirical models of the current spread in cortical tissue (*Tehovnik and Slocum, 2007*; *Winawer and Parvizi, 2016*; *Bosking et al., 2017*) to simulate the effects of stimulation current on the phosphene size. The accurate modeling of this biophysical relationship can help to increase the validity of simulation studies and brings fundamental SPV research closer to addressing questions regarding the practical real-life requirements of a visual prosthesis. Furthermore, the explicit link between the modeled area of cortical activation and the simulated phosphene sizes and locations makes our software suitable for including new receptive field modeling results (see *Figure 1—figure supplement 1* for an example simulation based on 3D receptive field modelling using third party software). Future studies that target other structures than V1 that contain a retinotopic map, such as the LGN, can also use the simulator by replacing the V1 map with a retinotopic map of the respective brain structure. Collaborative international projects such as the PRIMatE Resource Exchange (PRIME-RE) offer advanced tools which allow to fit probabilistic retinotopic maps generated from large samples to any individual NHP brain (*Klink et al., 2021*; *Messinger et al., 2021*) and it is currently possible to accurately predict human cortical receptive field mapping based on anatomical scans (*Benson et al., 2012*; *Benson et al., 2014*), or other retinotopic mapping strategies that do not rely on visual input (*Goebel et al., 2022*; *Bock et al., 2015*). This opens new doors for future research into the functionality of visual prostheses with limited visual field coverage. Thanks to the machine learning compatibility, the model can also be used for the pre-operative optimization of the implant placement and design (*van Hoof, 2022*).

### Threshold and brightness

The results presented in *Figures 2 and 3* indicate that our simulator closely models existing psychophysical data on the stimulation thresholds and phosphene brightness, for different electrical stimulation settings. Note that the effects found by *Fernández et al., 2021* (that were modeled by us) are consistent with findings by other studies, which report brighter phosphenes for higher stimulation strengths (*Schmidt et al., 1996*), and a lower stimulation efficiency (i.e. higher total charge thresholds) for longer pulse trains or higher pulse widths and frequencies (*Niketeghad et al., 2020*). While the simulator is developed for intracortical electrodes and stimulation amplitudes in the range of micro-amperes, there is a wide range of literature describing the use of electrodes which are placed on the cortical surface (e.g. *Brindley and Lewin, 1968*; *Dobelle and Mladejovsky, 1974*; *Girvin et al., 1979*; *Winawer and Parvizi, 2016*; *Bosking et al., 2017*; *Niketeghad et al., 2020*; *Beauchamp et al., 2020*). These electrodes require higher currents to elicit phosphenes (in the range of milli-amperes), but the mechanisms underlying the generation of phosphenes are presumably similar to those of intracortical electrodes. The results from *Figure 3—figure supplement 1*, suggest that the implemented thresholding model for intracortical stimulation also generalizes to surface stimulation. Moreover, our results are in line with other computational models for detection thresholds in the somatosensory cortex (*Fridman et al., 2010*; *Kim et al., 2017*). Our results indicate how a leaky integrator model and normally-distributed activation thresholds, provide a suitable approximation of the tissue activation in cortical prostheses. Note that alternative, more complex, models can possibly predict the psychometric data more accurately. However, most probably, this will entail a trade-off with the simplicity and modularity of the current simulator. Future research may further improve our understanding of the neural processing underlying the conscious perception of phosphenes, possibly borrowing insights from the domain of natural vision. More elaborate theories on this matter have been developed and tested in *van Vugt et al., 2018*. More specific limitations and suggestions for future adaptations are discussed below, in the section 'General limitations and future directions'.

### Temporal dynamics

The results presented in *Figure 4* reveal that the model accounts for experimental data on the accommodation in response to repeated stimulation in time periods up to 200 s. However, in contrast to the

findings by *Schmidt et al., 1996*, our simulator predicts a moderate recovery over the next 1000s. Although we cannot provide an explanation for this difference, the modeled recovery largely stays within the 95% confidence interval of the experimental data. Similar to the other components of our simulator, the memory trace was chosen as a basic, yet effective, model of neural habituation. Possibly, a more complex, non-linear, model can more accurately fit the neurophysiological data. However, the presented model has the benefit of simplicity (there are only three parameters). Also, note that there is still some ambivalence in the clinical data. In contrast to the findings by *Schmidt et al., 1996*, some studies have found accommodation over different time scales and sometimes no accommodation at all (*Bak et al., 1990*; *Fernández et al., 2021*; *Bartlett et al., 1977*; *Dobelle and Mladejovsky, 1974*). More research is required for a better understanding of the neural response after repeated or prolonged stimulation.

### Phosphene shape and appearance

The appearance of phosphenes in our simulation (white, round, soft dots of light) are largely in line with previous reports on intracortical stimulation (*Schmidt et al., 1996*; *Bak et al., 1990*; *Fernández et al., 2021*). However, more elongated shapes and more complex shapes have been reported as well (*Bosking et al., 2017*; *Winawer and Parvizi, 2016*; *Bak et al., 1990*). By using separately generated phosphene renderings, our simulator enables easy adjustments of the appearance of individual phosphenes. Additionally, we incorporated the possibility to change the default Gaussian blob appearance into Gabor patches with a specific frequency and orientation. Regarding the color of phosphenes, there is still some ambivalence in the reports, including descriptions of phosphene color ranging from black or white to different tones of color (*Tehovnik and Slocum, 2007*; *Tehovnik et al., 2009*; *Schmidt et al., 1996*). Notably, increasing the stimulation amplitudes can lead the appearance to shift from colored to yellowish or white (*Schmidt et al., 1996*). This effect may be explained by the increased current spread for higher stimulation amplitudes, which is predicted to span multiple cortical columns coding for different visual characteristics (e.g. orientation or color), thus giving rise to phosphenes with amalgamated features (*Tehovnik and Slocum, 2007*). Currently, the limited amount of systematic data render it difficult to enable more accurate simulations of the variability in phosphene appearance.

## End-to-end optimization

### Dynamic encoding

The results presented in *Figure 7* demonstrate that our proposed realistic phosphene simulator is well-suited for the dynamic optimization in an end-to-end architecture. Our proof-of-principle video-encoding experiments are the first to explicitly optimize the stimulation across the temporal domain. This provides a basis for the further exploration of computationally-optimized dynamic stimulation patterns. Dynamic optimization of the stimulation may be necessary to counteract unwanted effects such as response fading due to accommodation after repeated or prolonged stimulation (*Schmidt et al., 1996*), or delayed phosphene perception after stimulation on- and offset. The inclusion of a realistic simulator in the optimization pipeline enables researchers to exploit the optimal combination of stimulation parameters to obtain precise control over the required perception. Moreover, besides acquiring optimal control over the transfer function from stimulation to phosphenes, dynamic phosphene encoding could also prove useful to expand the encoded information along the temporal domain (*Beauchamp et al., 2020*). Although this was not in the scope of the current study, our software is well-suited for simulation experiments that further investigate dynamic stimulation. Note that there remain some challenging perceptual considerations for the design of useful dynamical stimulation patterns (for an excellent review on asynchronous stimulation in relation to flicker fusion, form vision, and apparent motion perception, please see *Moleirinho et al., 2021*).

### Constrained, efficient stimulation for natural stimuli

Our second optimization experiment addressed a more natural and realistic context. The results presented in *Figure 8* demonstrate that our simulator is well-suited for the optimization of prosthetic vision to natural stimuli and that it can be configured to comply with constraints of the stimulation protocol. Note that the quality of the reconstructions for the constrained version of the encoder indicate that the model can still find an efficient information encoding strategy using a limited set of stimulation amplitudes (10 discrete values between 0 and 128 µA). These results are in line with previous

results on constrained end-to-end optimization, indicating that task-relevant information can be maximized under sparsity constraints (*de Ruyter van Steveninck et al., 2022a*). While in the current experiments the stimulation amplitude is maximized for the individual electrodes, future studies could investigate other sparsity constraints, such as a maximum total charge delivered per second across all electrodes. Ultimately, a visual prosthesis may need to prioritize task-relevant information, rather than providing an accurate description of the visual surroundings. For this reason, in recent SPV research with sighted human observers much attention is devoted to semantic (boundary) segmentation for discriminating the important information from irrelevant background (*Sanchez-Garcia et al., 2020*; *Han et al., 2021*; *Rasla and Beyeler, 2022*). Note that the explored image processing strategies in these behavioral studies are equally compatible with the automated optimization through an end-to-end machine learning pipeline. Our experiments exemplify how supervision targets obtained from semantic segmentation data can be adopted to promote task-relevant information in the phosphene representation. Furthermore, in addition to reconstruction of the input or labeled targets, another recent study experimented with different decoding tasks, including more interactive, goal-driven tasks in virtual game environments (*Küçükoğlu et al., 2022*). Although these proof-of-principle results remain to be translated to real-world tasks and environments, they provide a valuable basis for further exploration. Computational optimization approaches can also aid in the development of safe stimulation protocols, because they allow a faster exploration of the large parameter space and enable task-driven optimization of image processing strategies (*Granley et al., 2022a*; *Fauvel and Chalk, 2022*; *White et al., 2019*; *Küçükoğlu et al., 2022*; *de Ruyter van Steveninck et al., 2022a*; *Ghaffari et al., 2021*). Ultimately, the development of task-relevant scene-processing algorithms will likely benefit both from computational optimization experiments as well as exploratory SPV studies with human observers. The results presented in *Figure 8—figure supplements 3 and 4* demonstrate that the end-to-end pipeline can also be used for low electrode counts and that exploratory experimentation with non-linear interaction models is possible. With the presented simulator we aim to contribute a flexible toolkit that can be tailored to specific use cases.

## Interpretability and perceptual correspondence

Besides the encoding efficiency (characterized by the computational decodability of task-relevant information), it is important to consider the subjective interpretablity of the simulated phosphene representation. In *Figure 8* it can be observed that the model has successfully learned to preserve correspondences between the phosphene representation and the input image in all of the training conditions. However, as a more formal analysis was outside the scope of this study, we did not further quantify the subjective interpretability. In our model the subjective interpretability was promoted through the regularization loss between the simulated phosphenes and the input image. Similarly, a recent study adapted an auto-encoder architecture designed to directly maximize the perceptual correspondence between a target representation and the simulated percept in a retinal prosthesis, using basic stimuli (*Granley et al., 2022a*). The preservation of subjective interpretability in an automated optimization pipeline remains a non-trivial challenge, especially when using natural stimuli. This may be even more important for cortical prostheses, as the distinct punctate phosphenes are in nature very dissimilar from natural images, possibly hampering perceptual similarity metrics that rely on low-level feature correspondence. Eventually, the functional quality of the artificial vision will not only depend on the correspondence between the visual environment and the phosphene encoding, but also on the implant recipient's ability to extract that information into a usable percept. The functional quality of end-to-end generated phosphene encodings in daily life tasks will need to be evaluated in future experiments. Regardless of the implementation, it will always be important to include human observers (both sighted experimental subjects and actual prosthetic implant users) in the optimization cycle to ensure subjective interpretability for the end user (*Fauvel and Chalk, 2022*; *Beyeler and Sanchez-Garcia, 2022*).

## General limitations and future directions
### Performance and hardware

There are some remaining practical limitations and challenges for future research. We identify three considerations related to the the performance of our model and the required hardware for implementation in an experimental setup. First, although our model runs in real-time and is faster than the

state-of-the art realistic simulation for retinal prostheses (*Beyeler et al., 2017*), there is a trade-off between speed and the memory demand. Therefore, for higher resolutions and larger numbers of phosphenes, future experimental research may need to adapt a simplified version of our model - although most of the simulation conditions can be run easily with common graphical cards. While several simulators exist for cortical prostheses that run in real-time without requiring a dedicated graphics card (*Li, 2013*; *Wong et al., 2010*), none of these incorporate current spread-based models of phosphene size, realistic stimulation parameter ranges or temporal dynamics. An effort can be made to balance more lightweight hardware requirements with more realistic phosphene characteristics. Second, a suggestion for follow-up research, is to combine our simulator with the latest developments in mixed reality (XR) to enable immersive simulation in virtual environments. More specifically, a convenient direction would be the implementation of our simulator using the Cg shader programming language for graphics processing, which is used in 3D game engines like Unreal Engine, or Unity 3D, as previously demonstrated for epiretinal simulations by *Thorn et al., 2020*. Third and lastly, future studies could explore the effects of using eye-tracking technology with our simulation software. Even after loss of vision, the brain integrates eye movements for the localization of visual stimuli (*Reuschel et al., 2012*), and in cortical prostheses the position of the artificially induced percept will shift along with eye movements (*Brindley and Lewin, 1968*; *Schmidt et al., 1996*). Therefore, in prostheses with a head-mounted camera, misalignment between the camera orientation and the pupillary axes can induce localization problems (*Caspi et al., 2018*; *Paraskevoudi and Pezaris, 2018*; *Sabbah et al., 2014*; *Schmidt et al., 1996*). Previous SPV studies have demonstrated that eye-tracking can be implemented to simulate the gaze-coupled perception of phosphenes (*Cha et al., 1992c*; *Sommerhalder et al., 2004*; *Dagnelie et al., 2006*; *McIntosh et al., 2013*; *Paraskevoudi and Pezaris, 2021*; *Rassia and Pezaris, 2018*; *Titchener et al., 2018*; *Srivastava et al., 2009*). Note that some of the cited studies implemented a simulation condition where not only the simulated phosphene locations, but also the stimulation protocol depended on the gaze direction. More specifically, instead of representing the head-centered camera input, the stimulation pattern was chosen to encode the external environment at the location where the gaze was directed. While further research is required, there is some preliminary evidence that such a gaze-contingent image processing can improve the functional and subjective quality of prosthetic vision (*Caspi et al., 2018*; *Paraskevoudi and Pezaris, 2021*; *Rassia and Pezaris, 2018*; *Titchener et al., 2018*). Some example videos of gaze-contingent simulated prosthetic vision can be retrieved from our repository here. Note that an eye-tracker will be required to produce gaze-contingent image processing in visual prostheses and there might be unforeseen complexities in the clinical implementation thereof. The study of oculomotor behavior in blind individuals (with or without a visual prosthesis) is still an ongoing line of research (*Caspi et al., 2018*; *Kwon et al., 2013*; *Sabbah et al., 2014*; *Hafed et al., 2016*).

## Complexity and realism of the simulation

There are some remaining challenges regarding the realistic simulation of the effects of neural stimulation. A complicating factor is that cortical neuroprostheses are still in the early stages of development. Neurostimulation hardware and stimulation protocols are continuously being improved (*Beauchamp et al., 2020*), and clinical trials with cortical visual neuroprostheses are often limited to small numbers of blind volunteers (*Fernández and Normann, 2017*; *Troyk, 2017*). Therefore, it is no surprise that the amount of data that is available at the present moment is limited, often open for multiple interpretations, and sometimes contains apparent contradictory information. Notably, the trade-off between model complexity and accurate psychophysical fits or predictions is a recurrent theme in the verification and validation of the components implemented in our simulator. Our approach aims to comprehensively integrate a set of biologically plausible models, while striking a balance between real-time performance, flexibility, and biological realism. Combining models of current spread and knowledge about the retinotopic organization of the visual cortex with psychophysics allows us to link the space of electrical stimulation parameters with clinical perceptual reports as well as physiological knowledge in the NHP and clinical literature. These design choices play a role in some of the potential limitations of our current simulator. Here, we name a few of the important limitations and some interesting directions for future research. First, in our simulator, phosphenes are only rendered when the activation is above threshold. This might be an inaccurate depiction of the perceptual experience of an implant user, and in reality the distinction may be less strict. The conscious perception of phosphenes requires

considerable training and the detection process is influenced by attention (*Fernández et al., 2021*). Although our implementation is effective for modeling the psychometric data, alternative implementations could also be considered. The perceptual effect of different simulated phosphene threshold implementations for sighted subjects remains to be evaluated in future SPV work. Second, the leaky integrator and the memory trace that are implemented in our simulator might be an oversimplified model of tissue activation in the visual cortex and some non-linear dynamics might be missed. For instance, all data used in this study to fit and validate the model used symmetric, biphasic pulse trains, while other pulse shapes might lead to different neural or behavioral responses (*Merrill et al., 2005*). Also, several studies reported that higher stimulation amplitudes may give rise to double phosphenes (*Brindley and Lewin, 1968*; *Schmidt et al., 1996*; *Dobelle and Mladejovsky, 1974*; *Oswalt et al., 2021*), or a reduction of phosphene brightness (*Schmidt et al., 1996*). Furthermore, in contrast to the assumptions of our model, interactions between simultaneous stimulation of multiple electrodes can have an effect on the phosphene size and sometimes lead to unexpected percepts (*Fernández et al., 2021*; *Dobelle and Mladejovsky, 1974*; *Bak et al., 1990*). Although our software supports basic exploratory experimentation of non-linear interactions (see *Figure 1—figure supplement 3*, *Figure 8—figure supplement 4*), by default, our simulator assumes independence between electrodes. Multi-phosphene percepts are modeled using linear summation of the independent percepts. These assumptions seem to hold for intracortical electrodes separated by more than 1 mm (*Ghose and Maunsell, 2012*), but may underestimate the complexities observed when electrodes are nearer. Further clinical and theoretical modeling work could help to improve our understanding of these non-linear dynamics. A third limitation is that our simulator currently only models responses of V1 stimulation. Future studies could explore the possible extension of modeling micro-stimulation of the LGN (*Pezaris and Reid, 2007*) and higher visual areas, such as V2, V3, V4, or inferotemporal cortex (IT). In previous NHP research, reliable phosphene thresholds could be obtained with the stimulation of the LGN, V1, V2, V3A, and middle temporal visual area (MT) (*Pezaris and Reid, 2007*; *Murphey and Maunsell, 2007*). Furthermore, IT stimulation has been shown to bias face perception (*Afraz et al., 2006*). Similar effects have been confirmed in human subjects, and previous work has demonstrated that electrical stimulation of higher-order visual areas can elicit a range of feature-specific percepts (*Murphey et al., 2009*; *Lee et al., 2000*; *Schalk et al., 2017*). Our simulator could be extended with maps of other visual areas with clear retinotopy, and an interesting direction for future research will be the implementation of feature-specific percepts, including texture, shape, and color.

## Conclusion

We present a framework for the biologically plausible simulation of phosphene vision. The simulator models psychophysical and neurophysiological findings in a wide array of experimental results. Its phenomenologically accurate simulations allow for the optimization of visual cortical prostheses in a manner that drastically narrows the gap between simulation and reality, compared to previous studies of simulated phosphene vision. It can operate in real-time, making it a viable option for behavioral experiments with sighted volunteers. Additionally, owing to the PyTorch implementation and the differentiable operations, it is a good choice for machine learning approaches to study and optimize phosphene vision. The modular design of the simulator allows for straightforward adaptation of novel insights and improved models of cortical activation. In summary, our open-source, biologically plausible phosphene simulator aims to provide an accessible bedrock software platform that fits the needs of fundamental, clinical, and computational vision scientists working on cortical neuroprosthetic vision. With this work, we aspire to contribute to increasing the field's translational impact.

## Acknowledgements

This work was supported by three grants of the Dutch Organization for Scientific Research (NWO): STW grant number P15-42 'NESTOR', ALW grant number 823-02-010 and Cross-over grant number 17619 'INTENSE' and grant number 024.005.022 'DBI2', a Gravitation program of the Dutch Ministry of Science, Education and Culture; the European Union's Horizon 2020 research and innovation programme: grant number 899287, 'NeuraViper'; the Human Brain Project, grant number 650003. We thank Xing Chen for help with the compilation and reviewing of relevant literature regarding phosphene perception.

# Additional information

## Funding

| Funder | Grant reference number | Author |
| --- | --- | --- |
| Nederlandse Organisatie voor Wetenschappelijk Onderzoek | P15-42 'NESTOR' | Pieter Roelfsema<br>Richard van Wezel<br>Marcel van Gerven |
| Nederlandse Organisatie voor Wetenschappelijk Onderzoek | 17619 'INTENSE' | Pieter Roelfsema<br>Yağmur Güçlütürk<br>Richard van Wezel<br>Marcel van Gerven |
| HORIZON EUROPE Excellent Science | 899287 'NeuraViper' | Bodo Rueckauer<br>Yağmur Güçlütürk<br>Pieter Roelfsema |
| Horizon 2020 Framework Programme | 650003 | Pieter Roelfsema |
| Nederlandse Organisatie voor Wetenschappelijk Onderzoek | 024.005.022 'DBI2' | Pieter Roelfsema<br>Marcel van Gerven |
| Nederlandse Organisatie voor Wetenschappelijk Onderzoek | 823-02-010 | Pieter Roelfsema |

The funders had no role in study design, data collection and interpretation, or the decision to submit the work for publication.

## Author contributions

Maureen van der Grinten, Conceptualization, Software, Formal analysis, Investigation, Methodology, Writing – original draft, Writing – review and editing; Jaap de Ruyter van Steveninck, Antonio Lozano, Conceptualization, Software, Formal analysis, Supervision, Investigation, Methodology, Writing – original draft, Writing – review and editing; Laura Pijnacker, Software, Investigation, Methodology; Bodo Rueckauer, Software, Supervision, Methodology, Writing – review and editing; Pieter Roelfsema, Supervision, Funding acquisition, Methodology, Writing – review and editing; Marcel van Gerven, Richard van Wezel, Conceptualization, Supervision, Funding acquisition, Methodology, Writing – review and editing; Umut Güçlü, Yağmur Güçlütürk, Conceptualization, Supervision, Funding acquisition, Methodology

## Author ORCIDs

Jaap de Ruyter van Steveninck ⓘ https://orcid.org/0000-0002-2711-0889
Antonio Lozano ⓘ https://orcid.org/0000-0003-4508-1484
Pieter Roelfsema ⓘ http://orcid.org/0000-0002-1625-0034
Marcel van Gerven ⓘ http://orcid.org/0000-0002-2206-9098

## Decision letter and Author response

Decision letter https://doi.org/10.7554/eLife.85812.sa1
Author response https://doi.org/10.7554/eLife.85812.sa2

# Additional files

## Supplementary files

• MDAR checklist

## Data availability

The source code of our simulator can be retrieved from GitHub (copy archived at *de Ruyter van Steveninck, 2024a*). It is licensed under the GNU General Public License. The latest stable release can be installed using pip: $ pip install dynaphos. The code and data that were used for experimental modelling and analysis in this paper can be retrieved from GitHub (copy archived at *de Ruyter van*

*Stevenick, 2024b*). The computational optimization experiments in this paper were run using an adapted version of a previously published end-to-end learning pipeline (*de Ruyter van Steveninck et al., 2022a*). The code can be retrieved from GitHub (copy archived at *de Ruyter van Steveninck, 2024c*).

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
