## [Editor Report]

This important study presents a simulator for prosthetic vision (with open source code) whose design is informed by previous psychophysical and neuroanatomical work. The simulation is convincing and demonstrates significant improvements over past visual prosthesis simulations. This work will be of interest to those investigating the impact of cortical stimulation on perception, particularly those developing visual prostheses.

---

## [Decision Letter]

**Decision letter after peer review:**

Thank you for submitting your article "Biologically plausible phosphene simulation for the differentiable optimization of visual cortical prostheses" for consideration by *eLife*. Your article has been reviewed by 3 peer reviewers, and the evaluation has been overseen by Chris Baker as the Reviewing Editor/Senior Editor. The following individual involved in review of your submission has agreed to reveal their identity: Michael P Barry (Reviewer #2).

Essential revisions:

The reviewers all think the work has great potential and will be useful to the field, but also highlight a number of major limitations. These are all laid out clearly in their individual comments (below).

In a revision, we would like to see the modeling work extended with a clear acknowledgement of some of the limitations. In particular, a revision should address concerns about:

1) Model validation with quantitative approaches.

2) Cortical folding.

3) Phosphene mapping.

4) Multiple electrode stimulation and electrode interactions.

We anticipate this will require substantial revisions to the modeling and the manuscript and not just a discussion of these issues.

*Reviewer #1 (Recommendations for the authors):*

The authors should clarify which data was used to fit the model and which data was used to test its predictive ability. As presented, it appears to be more of a descriptive "one size fits all" model than a predictive one that could be used to generalize to new patients and data. To that end, the paper should also make an effort to evaluate the model more thoroughly and more quantitatively.

The paper claims that based on CORTIVIS results, phosphenes are Gaussian blobs. However, this is true only for single-electrode percepts. Ref. 6 clearly states that multi-electrode stimulation does not produce a linear summation of Gaussian blobs. This is therefore quite a strong assumption of the model that needs to be clearly stated and discussed – it makes it unlikely that the results presented in Figure 8 would translate to real patients.

It is puzzling to me that the paper would place such an emphasis on the regularization loss, which was thought to promote "subjective interpretability", but then use a relative weighting of 0.999 for the reconstruction loss and only 0.001 for regularization. I suspect the low weight has to do with training instabilities, but with a relative weight of 0.001 it is hard to argue that this term had any influence on the results at all.

*Reviewer #3 (Recommendations for the authors):*

– The mapping algorithm for visual field to the cortical surface ignores cortical folding

– It appears the stimulation models also do not take cortical folding into account

– Given the authors' previous report in JoV, 2022, it isn't clear how much of this paper is an advance.

– While the authors repeatedly describe the advances that an end-to-end mode confers, the practicality of such a system is not discussed. In particular, end-to-end optimization can only demonstrate that the critical intermediate representation -- the phosphene brightnesses over time -- retains sufficient information for decoding an approximation of the original image. It says little about the hypothetical implant recipient's ability to extract that information into a usable percept.

– Figure 3 -- the dashed lines are difficult to visually parse.

– The simulator appears to assume perfect knowledge of phosphene location in the visual field; this assumption is implausible

– Some of the claims for performance are questionable: 10,000 phosphenes on an image of 64x64 pixels is not a reasonable assessment.

– Speed is good, but requires substantial hardware to achieve the claimed performance. Other, similar simulations from the literature report as-good or better performance without a multi-thousand dollar GPU. Although the published simulations do not include as detailed models as included here, the substantial difference gives one pause.

– Focusing on cortical stimulation leaves a significant portion of the literature unexplored, an unreasonable narrowing as the majority of simulation literature is fundamentally agnostic as to the targeted brain area, or at least adaptable to different areas with minimal effort, a claim that the authors here also make of their system.

[Editors' note: further revisions were suggested prior to acceptance, as described below.]

Thank you for resubmitting your work entitled "Towards biologically plausible phosphene simulation for the differentiable optimization of visual cortical prostheses" for further consideration by *eLife*. Your revised article has been evaluated by Chris Baker as Senior and Reviewing Editor.

The revised manuscript was re-evaluated by two of the initial reviewers. While the manuscript has been improved there are some remaining issues that need to be addressed, as outlined below:

*Reviewer #3 (Recommendations for the authors):*

Thank you for updating the manuscript and further demonstrating the capabilities of your simulator. The manuscript would be improved by addressing the below issues:

1) Overall, the manuscript should focus more on simulator demonstrations that reflect existing visual prosthetic technology, instead of highlighting examples with hundreds or thousands of noninteracting phosphenes in Figures 1 and 6-8. Examples using 10-60 phosphenes with nontrivial interactions should be prominent in the main manuscript. Even when only considering available channels, Fernández et al.'s system, as tested, could only stimulate up to 16 channels simultaneously, and the Orion device only has 60 channels. The ability to optimize hundreds of independent phosphenes will be very important in the future when devices are shown to be able to create that many phosphenes simultaneously. Until the field reaches that point, however, emphasizing examples with such large collections of phosphenes encourages misconceptions regarding the capabilities and existing challenges of visual prostheses.

2) The authors provide nice examples of how nonlinear interactions between single-electrode phosphenes can be rendered in Figure 1—figure supplement 3. While this is a nice demonstration, the authors should put more emphasis on this capability of the simulator. Alongside the end-to-end demonstrations of how the simulator performs assuming independent electrodes and phosphenes, the authors should include at least one end-to-end demonstration of how the simulator performs assuming nontrivial nonlinearities with fewer than 60 phosphenes. If such constraints appear to eliminate the meaningful utility of the simulator and its optimization process, the authors should thoroughly discuss this issue.

3) This simulator attempts to take numerous biological factors into account to translate electrode locations and stimulation parameters into simulated phosphenes, but also offers many points at which users can make manual adjustments. As the ideas of biological plausibility and simulator flexibility are both raised frequently, it would be good for the authors to specify what aspects of biological plausibility might be lost or maintained when users take advantage of different forms of flexibility in the simulator. For example, for the "most basic mode" of phosphene mapping described in lines 283-286, how much of the biological modeling is bypassed? Are V1 stimulation locations assumed based on the phosphene locations to calculate other phosphene characteristics?

4) Do the additions of nonlinear phosphene interactions, such as the ones in Figure 1—figure supplement 3, have any significant effect on simulator speed?

5) Aside from the instances of cross-validation, the authors frequently use the terms "validate" or "validation" when "verify" or "verification" would be more accurate. Particularly when the authors are demonstrating that model output reasonably matches the data for which it was configured to fit, the authors should not use the term validation.

6) The authors refer to cortical-surface electrodes generally as ECoG electrodes, but only a subset of the referenced studies used electrocorticography arrays. The Brindley and Lewin, Dobelle, and Orion systems did not record neural activity, and thus would not be classified as ECoG systems.

7) It would be useful for the authors to provide an example of how brightness accommodation is taken into account by the simulation over time.

8) In the right panel of Figure 2, it is unclear what "1e-7" at the top of the panel signifies.

9) It can be confusing how both memory trace and minimal input charge use the symbol Q.

10) In Figure 1—figure supplement 1, noise from a normal distribution with σ = 0.03 degrees is probably a bad example for representing uncertainty in phosphene location. Pointing responses from implantees can have standard deviations on the order of 3 degrees, so achieving a standard error of 0.03 degrees would require around 1000 localization trials per phosphene. Although this is just an example and the simulator can use any level of uncertainty, a more meaningful example of noise might use σ around 0.5-1.0 degrees.

11) Lines 141-142: The definition for stimulation threshold is vague. The threshold of perception is usually defined as the level at which the probability of stimulus detection is 50%, or sometimes 75%. Is there a specific probability associated with "reliably produce a visible phosphene"?

12) Line 270: The model is described as memory intensive. What ranges of memory are required?

13) Lines 323-329: Phosphene size is calculated based on the current provided, but not total charge. How is pulse width taken into account for phosphene sizes?

14) Lines 381-384: Activation thresholds are determined purely on a per-electrode basis. The authors should discuss how reduced charge-per-electrode thresholds with multi-electrode stimulation can be included in the simulator (e.g., less charge per electrode required when using 2 or 4 adjacent electrodes instead of a single electrode).

15) Line 453: The reference to the coefficient of determination should be paired with a clarification that this was only a verification of the parameter fitting process and not a demonstration of how the simulator matches unseen data.

16) Lines 758-759: The text mentions considerations for phosphene-perception delays after stimulation onset. Stimulation strategies will also be important for addressing phosphene perception persisting after stimulation offset. Is such undesired persistence modeled at all by the simulation?

17) Line 1023: The linked repository appears to be inaccessible: https://github.com/neuralcodinglab/dynaphos-experiments

---

## [Author Response]

Essential revisions:Reviewer #1 (Recommendations for the authors):The authors should clarify which data was used to fit the model and which data was used to test its predictive ability. As presented, it appears to be more of a descriptive "one size fits all" model than a predictive one that could be used to generalize to new patients and data. To that end, the paper should also make an effort to evaluate the model more thoroughly and more quantitatively.

As described in the public review response, we have adapted our model evaluations in several ways:

The fit of the thresholding mechanism was adapted to include a 3-fold cross validation, where part of the data was excluded during the fitting. The results of the cross-validation are now presented in panel D of Figure 3.For the components of the model calculating the phosphene size and brightness, the text was adapted to present these elements as fits and not predictions.To demonstrate model generalizability, the thresholding model was fit, using cross validation, to surface electrode data from several other clinical studies (Dobelle and Mladejovsky, 1974; Girvin et al., 1979; Niketeghad et al., 2019). Model predictions for these validation experiments are presented in Figure 3—figure supplement 1.

The paper claims that based on CORTIVIS results, phosphenes are Gaussian blobs. However, this is true only for single-electrode percepts. Ref. 6 clearly states that multi-electrode stimulation does not produce a linear summation of Gaussian blobs. This is therefore quite a strong assumption of the model that needs to be clearly stated and discussed – it makes it unlikely that the results presented in Figure 8 would translate to real patients.

We agree with the reviewer that the assumption of linear summation in our simulator does not correspond with all clinical observations in the literature: we now mention that linear summation is more likely when the distance between intracortical electrodes is larger than 1 mm (Ghose and Maunsell, 2012) and that the interference patterns are more likely for smaller distances.

Based on the reviewer’s suggestions, we have made several textual adjustments to improve on the clarity about our model assumptions and limitations. Furthermore, in Figure 1—figure supplement 2 and Figure 1—figure supplement 3 we now also demonstrate an example approach of how our simulator could be adapted to simulate arbitrary phosphene shapes and electrode interactions.

Adjustments:

Added Figure 1—figure supplement 2 on irregular phosphene perceptsLines 957-970: Furthermore, *in contrast to the assumptions of our model*, interactions between simultaneous stimulation of multiple electrodes can have an effect on the phosphene size and sometimes lead to unexpected percepts (Fernandez *et al.*, 2021, Dobelle and Mladejovsky 1974, Bak *et al.*, 1990). Although our software supports basic exploratory experimentation of non-linear interactions (see Figure 1—figure supplement 3), by default, our simulator assumes independence between electrodes. Multiphosphene percepts are modeled using linear summation of the independent percepts. These assumptions seem to hold for intracortical electrodes separated by more than 1 mm (Ghose and Maunsell, 2012), but may underestimate the complexities observed when electrodes are nearer. Further clinical and theoretical modeling work could help to improve our understanding of these non-linear dynamics.

It is puzzling to me that the paper would place such an emphasis on the regularization loss, which was thought to promote "subjective interpretability", but then use a relative weighting of 0.999 for the reconstruction loss and only 0.001 for regularization. I suspect the low weight has to do with training instabilities, but with a relative weight of 0.001 it is hard to argue that this term had any influence on the results at all.

We thank the reviewer for this remark. Mistakenly, we have reported the wrong numbers. The reconstruction loss and the regularization loss were weighted 0.00001 and 0.99999, respectively. Also, we have now specified that here the regularization loss is measured as the MSE between the brightness of the phosphene center pixels and the corresponding brightness in the input image. The high relative regularization loss ensures that the phosphenes look similar to the input image. Note that the regularization loss is calculated by comparing the input with the phosphene simulation and backpropagated through the encoder. Because only the encoder and not the decoder is affected by the regularization loss, the training of the decoder is driven entirely by the reconstruction loss. In summary:

The encoder is trained using a very small reconstruction loss component and a big regularization component to drive phosphene encodings that look similar to the input.The decoder is trained using the reconstruction component.

Accordingly, we have made the following textual adjustment:

Lines 578-586: We increased the training stability by taking the regularisation loss as the MSE between the pixel brightness at the phosphene centers and the corresponding pixel brightness in the input image. Furthermore, we adapted the relative weights of the reconstruction loss and the regularization loss to 0.00001 and 0.99999, respectively. Note that the regularization loss merely promotes similarity between the phosphene encoding and the input and the decoder is unaffected by the regularization loss.

Reviewer #3 (Recommendations for the authors):– The mapping algorithm for visual field to the cortical surface ignores cortical folding

Cortical folding is indeed an issue that cortical implants will need to consider. However, we chose to make the simulator as generally applicable as possible and the model can simulate any set of phosphene locations, also those produced when the calcarine sulcus is not implanted. As illustration we added a supplementary figure (Figure 1—figure supplement 1) demonstrating a feasible electrode grid placement on a 3D brain, generating the phosphene locations from estimated receptive field maps.

– It appears the stimulation models also do not take cortical folding into account

We thank the reviewer for this remark. We indeed did not include cortical folding into account in the stimulation models, as we assume that the area of activation will be on such a small scale that cortical folding will not have a notable impact (Ghose and Maunsell, 2012). However, such effects could be incorporated in a straightforward way to finetune the simulator to a specific patient’s perceptual experiences. Figure 1—figure supplement 3 displays an example where our code is finetuned to simulate irregular phosphene percepts,

– Given the authors' previous report in JoV, 2022, it isn't clear how much of this paper is an advance.

The improvements of our simulator compared to simulators that are presented in earlier studies are now described in the introduction.

The two studies had different aims: while the previous paper presents a pipeline for the endto-end optimization of phosphene encodings, our current study presents a simulator that can be used both in simulation experiments with human subjects and as well as in an existing end-to-end pipeline. We have improved the description of the differences between the two studies.

We have made the following textual changes:

Lines 232-253: Again, an important drawback is that current computational studies use simplified simulations of cortical prosthetic vision. This problem is addressed in the current study. Note that end-to-end machine learning pipelines rely on gradient propagation to update the parameters of the phosphene encoding model. Consequently, a crucial requirement is that the simulator makes use of differentiable operations to convert the stimulation parameters to an image of phosphenes – a requirement that is met by the proposed simulator. To evaluate the practical usability of our simulator in an end-to-end framework, we replicated and adapted the experiments by (de Ruyter van Steveninck, Güçlü et al., 2022), replacing their simulator with ours. The currently proposed simulator can be compared to the simulator that was used in the aforementioned work: Our simulator can handle temporal sequences and our experiments explore a more biologically grounded simulation of phosphene size and locations. Furthermore, instead of a more abstract or qualitative description of the required stimulation ('on' / 'off'), we included biologically inspired methods to model the perceptual effects of different stimulation parameters such as the current amplitude, the duration, the pulse width and the frequency.Lines 494-497: To validate that the simulator can conveniently be incorporated in a machine learning pipeline, we replicated an existing SPV pipeline by (de Ruyter van Steveninck, Güçlü et al., 2022), replacing the simulator with our biologically plausible simulator.

– While the authors repeatedly describe the advances that an end-to-end mode confers, the practicality of such a system is not discussed. In particular, end-to-end optimization can only demonstrate that the critical intermediate representation -- the phosphene brightnesses over time -- retains sufficient information for decoding an approximation of the original image. It says little about the hypothetical implant recipient's ability to extract that information into a usable percept.

The reviewer raises a valid point. On the one hand, prior literature (Granley *et al.*, 2022; Fauvel *et al.*, 2022; White *et al.*, 2019; Küçükoglü *et al.* 2022; de Ruyter van Steveninck, Güçlü *et al.*, 2022; Ghaffari *et al.*, 2021) indicate that computational optimization pipelines can be beneficial in general. On the other hand, simulation models differ from reality. The present study aims to reduce the gap between simulation models and the clinical reality. We do not yet know the experience of the end user. We now avoid claims about the benefits of end-to-end optimization, as the evaluation thereof is outside the scope of the current paper. We have now made these nuances more explicit and have further clarified the distinction between our own study aims and the hypothesized benefits stated in prior work.

Lines 842-853: Eventually, the functional quality of the artificial vision will not only depend on the correspondence between the visual environment and the phosphene encoding, but also on the implant recipient's ability to extract that information into a usable percept. The functional quality of end-to-end generated phosphene encodings in daily life tasks will need to be evaluated in future experiments. Regardless of the implementation, it will always be important to include human observers (both sighted experimental subjects and actual prosthetic implant users) in the optimization cycle to ensure subjective interpretability for the end user (Fauvel et al., 2022; Beyeler and Sanchez-Garcia, 2022).Lines 810-819: Computational optimization approaches can also aid in the development of safe stimulation protocols, because they allow a faster exploration of the large parameter space and enable task-driven optimization of image processing strategies (Granley et al., 2022; Fauvel et al., 2022; White et al., 2019; Küçükoglü et al. 2022; de Ruyter van Steveninck, Güçlü et al., 2022; Ghaffari et al., 2021). Ultimately, the development of taskrelevant scene-processing algorithms will likely benefit both from computational optimization experiments as well as exploratory SPV studies with human observers. With the presented simulator we aim to contribute a flexible toolkit for such experiments.

– Figure 3 -- the dashed lines are difficult to visually parse.

We have now removed the distinction between solid/dashed lines in the right panel, and the corresponding lines in the figure caption. (Note that this is now Figure 2 in the revised manuscript)

– The simulator appears to assume perfect knowledge of phosphene location in the visual field; this assumption is implausible

We agree that the assumption of having perfect knowledge of phosphene locations is implausible. For this reason, the mechanism for specifying electrode locations on the simplified cortical map includes noise. The simulator does, however, assume that phosphene locations are stable over time (e.g. Bak et al. 1990, Chen et al. 2020, Fernández et al. 2021). In that sense, the uncertainty of phosphene locations is taken into account once, at initialization of the simulator, and phosphene locations are predictable after this point. Furthermore, the simulator can be initialized with any arbitrary phosphene map, allowing researchers to include various levels of uncertainty in their simulations.

We adapted the description of the cortical mapping process to further clarify our assumptions regarding the sources of noise. Moreover, a supplementary figure has been added to showcase the possibility of using other methods for electrode and phosphene location initializations. The description of this figure includes an explicit mention of where in the pipeline uncertainty is assumed and noise added.

Lines 315-319: Moreover, to simulate imperfect knowledge of electrode or phosphene locations, and malfunctioning electrodes, the cortical mapping methods include parameters for the introduction of noise and electrode dropout.Figure 1—figure supplement 1 caption: To simulate imperfect knowledge of the electrode or phosphene locations, a small normally distributed noise (σ = 0.03°) was added to the determined receptive fields. Note that as the simulator is initialized with the phosphene locations in the visual field, any assumptions about feasible electrode locations, uncertainties and other sources of noise can be incorporated flexibly.

– Some of the claims for performance are questionable: 10,000 phosphenes on an image of 64x64 pixels is not a reasonable assessment.

We agree, our aim was to test the performance limits of the software and for completeness we evaluated all combinations of phosphene counts (between 10 and 10.000) and image resolutions (between 64x64 and 1024x1024). These permutations give insight in how the phosphene counts and image resolution impacts on the processing speed of the simulator. Our computational experiments simulated 1000 phosphenes and 256 x 256 pixels.

To avoid any confusion, we have now incorporated the following changes:

We have now moved the validation experiments (that evaluate the biological plausibility) to the beginning of the Results section as these are the most relevant results. The performance experiments (of secondary importance) are now presented afterwards.

We have added the following explanatory sentence to the figure caption (Figure 5 in the revised manuscript): “Note that these data are presented only for evaluating the software performance. For some combinations of phosphene count and image resolution (e.g. 10.000 phosphenes in a 64 x 64 image) there are fewer pixels then phosphenes”

– Speed is good, but requires substantial hardware to achieve the claimed performance. Other, similar simulations from the literature report as-good or better performance without a multi-thousand dollar GPU. Although the published simulations do not include as detailed models as included here, the substantial difference gives one pause.

Unquestionably, there are alternative strategies for the simulation of phosphene vision which can be run in real-time on a CPU. However, our simulator provides two important advantages that justify the requirement of parallel processing:

It has improved realism and biological plausibility. This is the most important motivation for our work.It allows the propagation of gradients, which enable the use in gradient-based optimization frameworks, including end-to-end experiments.

Note that the presented results in Figure 2 (index refers to the originally submitted manuscript) might have given the false impression that our simulator requires a multithousand dollar GPU. However, our simulator can also be run on a range of different devices, and the level of detail of the simulation is up to the user. See point 4 of the public responses: even on a commonly-used GPU that is priced around 600 euros our simulator runs detailed simulations with real-time performance.

Furthermore, GPUs are becoming more widespread because many behavioral and computational paradigms rely on them, for instance, to render virtual environments, or to train deep learning models.

We have incorporated the following textual adjustment:

Lines 860-873: Firstly, although our model runs in real-time and is faster than the state-of-the art realistic simulation for retinal prostheses (Beyeler et al., 2017), there is a tradeoff between speed and the memory demand. Therefore, for higher resolutions and larger number of phosphenes, future experimental research may need to adapt a simplified version of our model – although most of the simulation conditions can be run easily with common graphical cards. While several simulators exist for cortical prostheses that run in real time without requiring a dedicated graphics card (e.g. Li et al., 2013; Fehervari et al., 2010), none of these incorporate current spread-based models of phosphene size, realistic stimulation parameter ranges or temporal dynamics. An effort can be made to balance more lightweight hardware requirements with more realistic phosphene characteristics.

– Focusing on cortical stimulation leaves a significant portion of the literature unexplored, an unreasonable narrowing as the majority of simulation literature is fundamentally agnostic as to the targeted brain area, or at least adaptable to different areas with minimal effort, a claim that the authors here also make of their system.

We agree, and the simulator could also be adapted to emulate thalamic stimulation (Pezaris and Reid, 2007). However, retinal stimulation has been shown to produce distinct visual percepts due to the organization of axonal pathways, and the modeling and simulation of these processes is well addressed by Beyeler et al. (2017). We are unsure about optic nerve stimulation, because it may produce percepts with a different dependence on stimulation parameters because the location of phosphenes changes if stimulation frequency and amplitude is varied (Delbeke et al. 2003).

In addition, following the reviewer's advice, we have now explored the possibility of fitting our simulator to results from the clinical literature on cortical surface electrodes. The results are presented in Figure 3—figure supplement 1.

Textual additions:

Lines 107-110: The modular design of the simulator allows for future extensions to simulate brain stimulation in other regions such as lateral geniculate nucleus (LGN) or higher visual areas (Murphey et al., 2007; Pezaris and Reid, 2007; Panetsos et al., 2011; Mirochnik and Pezaris, 2019)Lines 319-322: We note, however, that the framework is compatible with the retinotopic maps of other structures, such as the LGN, which is the structure providing input to the primary visual cortex.Lines 649-652: Future studies that target other structures than V1 that contain a retinotopic map, such as the LGN, can also use the simulator by replacing the V1 map with a retinotopic map of the respective brain structure.

[Editors’ note: what follows is the authors’ response to the second round of review.]

The revised manuscript was re-evaluated by two of the initial reviewers. While the manuscript has been improved there are some remaining issues that need to be addressed, as outlined below:Reviewer #3 (Recommendations for the authors):Thank you for updating the manuscript and further demonstrating the capabilities of your simulator. The manuscript would be improved by addressing the below issues:1) Overall, the manuscript should focus more on simulator demonstrations that reflect existing visual prosthetic technology, instead of highlighting examples with hundreds or thousands of noninteracting phosphenes in Figures 1 and 6-8. Examples using 10-60 phosphenes with nontrivial interactions should be prominent in the main manuscript. Even when only considering available channels, Fernández et al.'s system, as tested, could only stimulate up to 16 channels simultaneously, and the Orion device only has 60 channels. The ability to optimize hundreds of independent phosphenes will be very important in the future when devices are shown to be able to create that many phosphenes simultaneously. Until the field reaches that point, however, emphasizing examples with such large collections of phosphenes encourages misconceptions regarding the capabilities and existing challenges of visual prostheses.

Indeed, many of the demonstrations in our study focus on the ability to optimize hundreds of independent phosphenes. This is an important future challenge that requires further exploration, especially given recent advances demonstrating higher channel-count visual prostheses (see Chen et. 2020, Science). Nevertheless, we fully agree with the reviewer on the importance of addressing limitations of existing contemporary devices. Depending on the requirements of the user, our simulator can be configured to simulate low-channel-count prostheses just as well as high-channelcount prostheses.

We thank the reviewer for the concrete suggestions. To demonstrate that our simulator is not limited to large electrode counts and can simulate low-resolution vision of contemporary prostheses, we have added a supplementary experiment with only 60 available electrodes.

**Author response image 1. sa2fig1:** 

Furthermore, we implemented a basic interaction model in the end-to-end pipeline. The results are visualized in Figure 8—figure supplement 3 and Figure 8—figure supplement 4.

2) The authors provide nice examples of how nonlinear interactions between single-electrode phosphenes can be rendered in Figure 1—figure supplement 3. While this is a nice demonstration, the authors should put more emphasis on this capability of the simulator. Alongside the end-to-end demonstrations of how the simulator performs assuming independent electrodes and phosphenes, the authors should include at least one end-to-end demonstration of how the simulator performs assuming nontrivial nonlinearities with fewer than 60 phosphenes. If such constraints appear to eliminate the meaningful utility of the simulator and its optimization process, the authors should thoroughly discuss this issue.

We thank the reviewer for the suggestion of putting extra emphasis on nonlinear interactions, and point back to the response on point 1. The presented additional experiments demonstrate that the exploration of stimulation effects with custom nonlinear electrode interactions is possible.

3) This simulator attempts to take numerous biological factors into account to translate electrode locations and stimulation parameters into simulated phosphenes, but also offers many points at which users can make manual adjustments. As the ideas of biological plausibility and simulator flexibility are both raised frequently, it would be good for the authors to specify what aspects of biological plausibility might be lost or maintained when users take advantage of different forms of flexibility in the simulator. For example, for the "most basic mode" of phosphene mapping described in lines 283-286, how much of the biological modeling is bypassed? Are V1 stimulation locations assumed based on the phosphene locations to calculate other phosphene characteristics?

The flexibility of the simulator lies in the modular code implementation, and it is up to the user to remove or reconfigure specific modeling features for certain goals. However, for the work presented in the current manuscript, we always used the entire simulator’s features including all the biologically plausible modules.

We believe that the phrasing ‘in the most basic mode’ may have been a bit misleading, since it suggests that the simulator uses several predefined modes. What we meant to say was: users of the simulator can provide a list of electrode locations or phosphene locations to the simulator to base the simulations on clinical data. We also provide the code for generating a random list of phosphene locations based on equally-distant electrode locations on a flattened cortical map of V1.

The phrasing has been adapted in the manuscript for clarification.

4) Do the additions of nonlinear phosphene interactions, such as the ones in Figure 1—figure supplement 3, have any significant effect on simulator speed?

When measuring timings, we found that calculating the interactions in figure 3 took 0.02ms on average, thus the impact on the simulation speed can be considered neglectable.

5) Aside from the instances of cross-validation, the authors frequently use the terms "validate" or "validation" when "verify" or "verification" would be more accurate. Particularly when the authors are demonstrating that model output reasonably matches the data for which it was configured to fit, the authors should not use the term validation.

We thank the reviewer for the suggestion to clarify the text, and we have added the term ‘verification’ at several locations in the text where validation was mentioned, to indicate that not all parts of the model were validated in the strict sense of the word.

6) The authors refer to cortical-surface electrodes generally as ECoG electrodes, but only a subset of the referenced studies used electrocorticography arrays. The Brindley and Lewin, Dobelle, and Orion systems did not record neural activity, and thus would not be classified as ECoG systems.

We have adapted the text to clarify that the electrodes described are placed on the cortical surface (as opposed to intracortical electrodes), and we have removed the

“ECoG” indication.

7) It would be useful for the authors to provide an example of how brightness accommodation is taken into account by the simulation over time.

We agree that such an example would be quite useful to the reader. We have added Figure 4—figure supplement 1 as an example showcasing the effects of brightness accommodation as well as delayed phosphene onset in response to 7 seconds of continuous stimulation.

8) In the right panel of Figure 2, it is unclear what "1e-7" at the top of the panel signifies.

“1e-7” has been changed to “10^-7” to clarify the range of the activation values in this figure.

9) It can be confusing how both memory trace and minimal input charge use the symbol Q.

We agree, and we have therefore changed the symbol for the memory trace to B.

10) In Figure 1—figure supplement 1, noise from a normal distribution with σ = 0.03 degrees is probably a bad example for representing uncertainty in phosphene location. Pointing responses from implantees can have standard deviations on the order of 3 degrees, so achieving a standard error of 0.03 degrees would require around 1000 localization trials per phosphene. Although this is just an example and the simulator can use any level of uncertainty, a more meaningful example of noise might use σ around 0.5-1.0 degrees.

We appreciate the reviewer's remark indicating that uncertainty in the experimental estimation of phosphene locations can be of a higher magnitude as the one used in our paper. However, the mentioned pointing error reflects a *reporting error* rather than the *perceptual spread* of phosphene locations. Our simulator simulates the *perceptual experience* of the implant user, and incorporates uncertainty and noise due to implantation inaccuracies and visuotopic map irregularities, and not *reporting inaccuracies*.

In other words: the simulated phosphene image in Figure 1—figure supplement 1 shows what the user would *see* as in contrast to what the person would manually *report*. Therefore, we disagree with the reviewer that increasing the noise level would give a more meaningful example. As such, we have chosen to not change the figure.

Note that the noise levels can be easily adjusted depending on the users requirements. An illustration of the result of setting the noise to 0.5 degrees is included here:

11) Lines 141-142: The definition for stimulation threshold is vague. The threshold of perception is usually defined as the level at which the probability of stimulus detection is 50%, or sometimes 75%. Is there a specific probability associated with "reliably produce a visible phosphene"?

We agree that this definition is imprecise and have changed the text to “produce a visible phosphene 50% of the time”.

12) Line 270: The model is described as memory intensive. What ranges of memory are required?

We have added the memory size (24GB) of the used GPU in the caption of figure 5. Note that simulations up to 265x265 pixels with up to 1000 phosphenes are also possible on relatively basic GPUs with a memory size of 2GB.

13) Lines 323-329: Phosphene size is calculated based on the current provided, but not total charge. How is pulse width taken into account for phosphene sizes?

Pulse width is not taken into account, as we did not find quantitative data describing the effects of either pulse width or frequency on the size of phosphenes, from intracortical electrodes trials. We made the choice to use the models described by Bosking and Tehovnik that are based only on stimulation current amplitude. We hope that more quantitative experimental data on phosphene size and the effects of several stimulation parameters become available, so that it will be possible to adapt the model to include these.

14) Lines 381-384: Activation thresholds are determined purely on a per-electrode basis. The authors should discuss how reduced charge-per-electrode thresholds with multi-electrode stimulation can be included in the simulator (e.g., less charge per electrode required when using 2 or 4 adjacent electrodes instead of a single electrode).

See the response on point 1. By default our simulator does not model these electrode interactions, as there is no sufficient literature on the underlying mechanistic models. However, we demonstrate that custom interaction models can be easily added (see Figure 1—figure supplement 2; Figure 1—figure supplement 3; Section ‘Small electrode counts and interactions’ ; Figure 8—figure supplement 4).

15) Line 453: The reference to the coefficient of determination should be paired with a clarification that this was only a verification of the parameter fitting process and not a demonstration of how the simulator matches unseen data.

We thank the reviewer for the suggestion: such a clarification has been added.

16) Lines 758-759: The text mentions considerations for phosphene-perception delays after stimulation onset. Stimulation strategies will also be important for addressing phosphene perception persisting after stimulation offset. Is such undesired persistence modeled at all by the simulation?

The leaky integrator model as described in the Materials and methods section implements both the delayed on- and offset. A sentence has been added in this section to clarify this: “By integrating over time, this model additionally implements the delayed on- and offset as described by several studies (38,39).” Furthermore, the right panel in Figure 2 displays the delayed phosphene offset of the activation and brightness after stimulation offset.

17) Line 1023: The linked repository appears to be inaccessible: https://github.com/neuralcodinglab/dynaphos-experiments

We thank the reviewer for this notification. We will make the repository publicly available upon publication of the manuscript. Meanwhile, all code that was used for the experiments is shared in the submission portal in order to make it accessible for the reviewers.